# GloptiNets: Scalable Non-Convex Optimization with Certificates

**Gaspard Beugnot**
gaspard.beugnot@inria.fr
Inria, École normale supérieure, CNRS, PSL Research University, 75005 Paris, France

**Julien Mairal**
julien.mairal@inria.fr
Univ. Grenoble Alpes, Inria, CNRS, Grenoble INP, LJK, 38000 Grenoble, France

**Alessandro Rudi**
alessandro.rudi@inria.fr
Inria, École normale supérieure, CNRS, PSL Research University, 75005 Paris, France

## Abstract

We present a novel approach to non-convex optimization with certificates, which handles smooth functions on the hypercube or on the torus. Unlike traditional methods that rely on algebraic properties, our algorithm exploits the regularity of the target function intrinsic in the decay of its Fourier spectrum. By defining a tractable family of models, we allow *at the same time* to obtain precise certificates and to leverage the advanced and powerful computational techniques developed to optimize neural networks. In this way the scalability of our approach is naturally enhanced by parallel computing with GPUs. Our approach, when applied to the case of polynomials of moderate dimensions but with thousands of coefficients, outperforms the state-of-the-art optimization methods with certificates, as the ones based on Lasserre's hierarchy, addressing problems intractable for the competitors.

## 1 Introduction

Non-convex optimization is a difficult and crucial task. In this paper, we aim at optimizing globally a non-convex function defined on the hypercube, by providing a certificate of optimality on the resulting solution. Let $h$ be a smooth function on $[-1, 1]^d$. Here we provide an algorithm that given $\widehat{x}$, an estimate of the minimizer $x_\star$ of $h$

$$x_\star = \underset{x \in [-1,1]^d}{\arg\min} \, h(x),$$

produces an $\epsilon$, that constitutes an explicit *certificate* for the quality of $\widehat{x}$, of the form

$$|h(x_\star) - h(\widehat{x})| \leq \epsilon_\delta,$$

with probability $1 - \delta$. The literature abounds of algorithms to optimize non-convex functions. Typically they are either *(a)* heuristics, very smart, but with no guarantees of global convergence Moscato et al. [1989], Horst and Pardalos [2013] *(b)* variation of algorithms used in convex optimization, which can guarantee convergence only to *local* minima Boyd and Vandenberghe [2004] *(c)* algorithms with only asymptotic guarantees of convergence to a global minimum, but no explicit certificates Van Laarhoven et al. [1987]. In general, the methods recalled above are quite fast to produce some

solution, but don't provide guarantees on its quality, with the result that the produced point can be arbitrarily far from the optimum, so they are used typically where non-reliable results can be accepted.

On the contrary, there are contexts where an explicit quantification of the incurred error is crucial for the task at hand (finance, engineering, scientific validation, safety-critical scenarios Lasserre [2009]). In these cases, more expensive methods that provide certificates are used, such as *polynomial sum-of-squares* (poly-SoS) Lasserre [2001, 2009]. These kinds of techniques are quite powerful since they provide certificates in the form above, often with machine-precision error. However, *(a)* they have reduced applicability since $h$ must be a multivariate polynomial (possibly sparse, low-degree) and must be known in its analytical form *(b)* the resulting algorithm is a semi-definite programming optimization on matrices whose size grows very fast with the number of variables and the degree of the polynomial, becoming intractable already in moderate dimensions and degrees.

Our approach builds and extends the more recent line of works on *kernel sum-of-squares*, and in particular the work of Woodworth et al. [2022] based on the Fourier analysis. It mitigates the limitations of poly-SoS methods in both aspects: *(a)* we can deal with any function $h$ (not necessarily a polynomial) for which the Fourier transform is known and *(b)* the resulting algorithm leverages the smoothness properties of the objective function as Woodworth et al. [2022] rather than relying on its algebraic structure leading to way more compact representations than poly-SoS. Contrary to Woodworth et al. [2022], we fully leverage the power of the certificate allowing for a drastic reduction of the computational cost of the method. Indeed, we cast the minimization in terms of a way smaller problem, similar to the optimization of a small neural network that, albeit again non-convex, produces efficiently a good candidate on which we then compute the certificate.

Notably, our focus lies on a posteriori guarantees: we define a family of models that allow for efficient computation of certificates. Once the model structure is established, we have ample flexibility in training the model, offering various possibilities to achieve good certificates in practical scenarios, while still using well-established and effective techniques in the field of deep neural networks (DNN) Goodfellow et al. [2016] to reduce the computational burden of the approach.

Our contributions can be summarized as follows:

- We propose a new approach to global optimization *with certificates* which drastically extends the applicability domain allowed by the state of the art, since it can be applied to any function for which we can compute the Fourier transform (not just polynomials).

- The proposed approach is naturally tailored for GPU computations and provides a refined control of time and memory requirements of the proposed algorithm, contrary to poly-SoS methods (whose complexity scales dramatically and in a rigid way with dimension and degree of the polynomial).

- From a technical viewpoint, we improve the results in Woodworth et al. [2022], by developing a fast stochastic approach to recover the certificate in high probability (theorem 3), and we generalize the formulation of the problem to allow the use of powerful techniques from DNN, still providing a certificate on the result (section 3, in particular alg. 1)

- In practical applications, we are able to provide certificates for functions in moderate dimensions, which surpasses the capabilities of current state-of-the-art techniques. Specifically, as shown in the experiments we can handle polynomials with thousands of coefficients. This achievement marks an important milestone towards utilizing these models to provide certificates for more challenging real-life problems.

## 1.1 Previous work

**Polynomial SoS.** In the field of certificate-based polynomial optimization, Lasserre's hierarchy plays a pivotal role Lasserre [2001, 2009]. This hierarchy employs a sequence of SDP relaxations with increasing size proportional to $O(r^d)$ (where $d$ is the dimension of the space and $r$ is a parameter that upper bounds the degree of the polynomial) and that ultimately converges to the optimal solution when $r \to \infty$. While Lasserre's hierarchy is primarily associated with polynomial optimization, its applicability extends beyond this domain. It offers a specific formulation for the more general moment problem, enabling a wide range of applications; see Henrion et al. [2020] for an introduction. For polynomial optimization problems such as in eq. (1), a significant amount of research has been dedicated to leveraging problem structure to improve the scalability of the hierarchy. This research

has predominantly focused on exploiting very specific sparsity patterns among the variables of the polynomial, enabling the handling in these restricted scenarios of instances ranging from a few variables to even thousands of variables Waki et al. [2006], Wang et al. [2021b,a]. There have been theoretical results regarding optimization on the hypercube Bach and Rudi [2023], Laurent and Slot [2022], but there are no algorithms handling them natively. Furthermore, alternative approaches exist that exploit different types of structure, such as the constant trace property Mai et al. [2022].

**Kernel SoS.** Kernel Sum of Squares (K-SoS) is an emerging research field that originated from the introduction of a novel parametrization for positive functions in Marteau-Ferey et al. [2020]. This approach has found application in various domains, including Optimal Control Berthier et al. [2022], Optimal Transport Muzellec et al. [2021] and modeling probability distribution Rudi and Ciliberto [2021]. In the context of function optimization, two types of theoretical results have been explored: *a priori* guarantees Rudi et al. [2020] and *a posteriori* guarantees Woodworth et al. [2022]. A priori guarantees offer insights into the convergence rate towards a global optimum of the function, giving a rate on the number of parameters and the complexity necessary to optimize a function up to a given error. For example, Rudi et al. [2020] proposes a general technique to achieve the global optimum, with error $\epsilon$ of a function that is $s$-times differentiable, by requiring a number of parameters essentially in the order of $O(\epsilon^{-s/d})$, allowing to avoid the curse of dimensionality in the rate, when the function is very regular, i.e., $s \geq d$, while typical black-box optimization algorithms have a complexity that scales as $\epsilon^{-d}$. A-posteriori guarantees focus on providing a certificate for the minimum found by the algorithm. In particular, Woodworth et al. [2022], provides both a-priori guarantee and a-posteriori certificates; however, the model considered makes it computationally infeasible to provide certificates in dimension larger than 2.

To conclude, approaches based on kernel-SoS allow to extend the applicability of global optimization with certificates methods to a wider family of functions and on exploiting finer regularity properties beyond just the number of variables and the degrees of a polynomial. By comparison, we focus on making the optimization amenable to high-performance GPU computation while retaining an a posteriori certificate of optimality.

## 2    Computing certificates with extended k-SoS

Without loss of generality (see next remark), with the goal of simplifying the analysis and using powerful tools from harmonic analysis, we cast the problem in terms of minimization of a *periodic* function $f$ over the torus, $[0, 1]^d$ (we will denote it also as $\mathbb{T}^d$). In particular, we are interested in minimizing periodic functions for which we know (or we can easily compute) the coefficients of its Fourier representation, i.e.

$$f_\star = \min_{z \in \mathbb{T}^d} f(z), \qquad f(z) = \sum_{\omega \in \mathbb{Z}^d} \widehat{f}_\omega e^{2\pi \mathrm{i} \omega \cdot z}, \quad \forall z \in \mathbb{T}^d, \tag{1}$$

where $\mathbb{Z}$ is the set of integers. This setting is already interesting on its own, as it encompasses a large class of smooth functions. It includes notably trigonometric polynomials, *i.e.* functions which have only a finite number of non-zero Fourier coefficients $\widehat{f}_\omega$. Optimization of trigonometric polynomials arises in multiple research areas, such as the optimal power flow Van Hentenryck [18] or quantum mechanics Hilling and Sudbery [2010]. Note that this problem is already NP-hard, as it encompasses for instance the Max-Cut problem Waldspurger et al. [2013]. Even so, we will consider the more general case where we can evaluate function values of $f$, along with its Fourier coefficient $\widehat{f}_\omega$, and we have access to its norm in a certain Hilbert space. This norm can be computed numerically for trigonometric polynomials, and more generally reflects the regularity (degree of differentiability) of the function, and thus the difficulty of the problem.

**Remark 1** (No loss of generality in working on the torus). *Given a (non-periodic) function* $h : [-1, 1]^d \to \mathbb{R}$ *we can obtain a periodic function whose minimum is exactly* $h_*$ *and from which we can recover* $x_\star$. *Indeed, following the classical Chebychev construction, define* $\cos(2\pi z)$ *as the componentwise application of* $\cos$ *to the elements of* $2\pi z$, *i.e.* $\cos(2\pi z) := (\cos(2\pi z_1), \ldots, \cos(2\pi z_d))$ *and define* $f$ *as* $f(z) := h(\cos(2\pi z))$ *for* $z \in [0, 1]^d$. *It is immediate to see that* (a) $f$ *is periodic, and,* (b) *since* $\cos(2\pi z)$ *is invertible on* $[0, 1]^d$ *and its image is exactly* $[-1, 1]^d$, *we have* $h_* = h(x_\star) = f(z_\star)$ *where*

$$x_\star = \cos(2\pi z_\star), \quad and \quad z_\star = \min_{z \in \mathbb{T}^d} f(z).$$

*We discuss an efficient representation of these problems in section 3.3.*

## 2.1 Certificates for global optimization and k-SoS

A general "recipe" for obtaining a certificates was developed in Woodworth et al. [2022] where, in particular, it was derived the following bound [Woodworth et al., 2022, see Thm. 2]

$$f_\star \geq \sup_{c \in \mathbb{R}, \, g \in \mathcal{G}_+} c - \|f - c - g\|_F \quad , \tag{2}$$

where $\|u\|_F$ is the $\ell_1$ norm of the Fourier coefficients of a periodic function $u$, i.e.

$$\|u\|_F := \sum_{\omega \in \mathbb{Z}^d} |\widehat{u}_\omega| \,, \tag{3}$$

and the sup is taken over $\mathcal{G}_+$ that is a class of non-negative functions. The paper Woodworth et al. [2022] then chooses $\mathcal{G}_+$ to be the set of *positive semidefinite models*, leading to a possibly expensive convex SDP problem. Our approach instead starts from the following two observations: *(a)* the lower bound in eq. (2) holds for any set $\mathcal{G}_+$ of non-negative functions, *not necessarily convex*, moreover *(b)* any candidate solution $(g, c)$ of the supremum in eq. (2) would constitute a lower bound for $f_\star$, so there is no need to solve eq. (2) exactly. This yields the following theorem

**Theorem 1.** *Given a point $\widehat{x} \in \mathbb{T}^d$ and a non-negative and periodic function $g_0 : \mathbb{T}^d \to \mathbb{R}_+$, we have*

$$|f(\widehat{x}) - f(x_\star)| \leq \|f - f(\widehat{x}) - g_0\|_F \tag{4}$$

*Proof.* Since $x_\star$ is the minimizer of $f$, then $f(x_\star) \leq f(\widehat{x})$. Moreover, since $c_0 = f(\widehat{x})$ and $g_0$ are feasible solutions for the r.h.s. of eq. (2), we have

$$f(\widehat{x}) \geq f(x_\star) \geq \sup_{c \in \mathbb{R}, \, g \in \mathcal{G}_+} c - \|f - c - g\|_F \geq c_0 - \|f - c_0 - g_0\|_F \,,$$

from which we derive that $0 \leq f(\widehat{x}) - f(x_\star) \leq \|f - f(\widehat{x}) - g_0\|_F$. $\qquad\square$

In particular, since any good candidate $g_0$ is enough to produce a certificate, we consider the following class of non-negative functions that can be seen as a *two-layer neural network*.

**Definition 1** (extended K-SoS model on the torus). *Let $K : \mathbb{T}^d \times \mathbb{T}^d \to \mathbb{R}$ be a periodic function in the first variable and let $m, r \in \mathbb{N}$. Given a set of anchors $\mathbf{Z} = (\mathbf{z}_1, \ldots, \mathbf{z_m}) \subset \mathbb{T}^d$ and a matrix $R \in \mathbb{R}^{r \times m}$, we define the K-SoS model $g$ with*

$$\forall \mathbf{x} \in \mathbb{T}^d, \quad g(\mathbf{x}) = \|R K_{\mathbf{Z}}(\mathbf{x})\|_2^2, \quad and \quad K_{\mathbf{Z}}(\mathbf{x}) = (K(\mathbf{x}, \mathbf{z}_1), \ldots, K(\mathbf{x}, \mathbf{z}_m)) \in \mathbb{R}^m. \tag{5}$$

The functions represented by the model above are non-negative and periodic. The model is an extension of the k-SoS model presented in Marteau-Ferey et al. [2020], where the points $(\mathbf{z}_1, \ldots, \mathbf{z_m})$ cannot be optimized. Moreover it has the following benefits at the expense of the convexity in the parameters:

1. The extended k-SoS models benefit of the good approximation properties of k-SoS models described in Marteau-Ferey et al. [2020] and especially Rudi and Ciliberto [2021], since they are a super-set of the k-SoS, that have optimal approximation properties for non-negative functions.

2. The extended model can have a *reduced number of parameters*, by choosing a matrix $R$ with $r = 1$ or $r \ll m$. This will drastically improve the cost of the optimization, while not impacting the approximation properties of the model, since a good approximation is still possible with already $r$ proportional to $d$ [Rudi et al., 2020, see Thm. 3].

3. The extended model *does not require any positive semidefinite constraint* on the matrix (contrary to the base model) that is typically a well-known bottleneck to scale up the optimization in the number of parameters Marteau-Ferey et al. [2020]. In the extended model we trade the positive semidefinite constraint with non-convexity. However this allows us to use all the advanced and effective techniques we know for unconstrained (or box-constrained) non-convex optimization for (two-layers) neural networks Goodfellow et al. [2016].

To conclude the picture on the k-SoS models, a critical aspect of the model is the choice of $K$, since it must guarantee good approximation properties and at the same time we need to compute easily its Fourier coefficients since we need to evaluate $\|f - c - g\|_F$. To this aim, a good candidate for $K$ are the *reproducing kernels* defined on the torus Steinwart and Christmann [2008]. We use shift-invariant kernels, enabling a convenient analysis of the associated RKHS through their Fourier Transform.

**Definition 2** (Reproducing kernel on the torus). *Let $q$ be a real function on $\mathbb{T}^d$, with positive Fourier Transform and $q(0) = 1$. Let $K$ be the kernel defined with*

$$\forall x, y \in \mathbb{T}^d, \quad K(x, y) = q(x - y) = \sum_{\omega \in \mathbb{Z}^d} \widehat{q}_\omega e^{2\pi \mathrm{i}\omega \cdot (x - y)}. \tag{6}$$

*Then, $K$ is a r.k bounded by 1. We denote $\mathcal{H}$ its Reproducing kernel Hilbert Space (RKHS) and by $\|\cdot\|_{\mathcal{H}}$ the associated RKHS norm*

$$\|f\|_{\mathcal{H}}^2 = \sum_{\omega \in \mathbb{Z}^d} |\widehat{f}_\omega|^2 / \widehat{q}_\omega.$$

*Define $\lambda(x) = q(x)^2$. We assume that we can* compute *(and* sample *from, see next section)* $\widehat{\lambda}_\omega$*, i.e., the Fourier transform of $\lambda$, corresponding to $(\widehat{q} \star \widehat{q})_\omega$, for all $\omega \in \mathbb{Z}^d$.*

By choosing such a $K$, the models inherit the good approximation properties derived in Marteau-Ferey et al. [2020], Rudi and Ciliberto [2021]. We conclude by recalling that shift-invariant r.k kernel have a positive Fourier transform due to Bochner's theorem Rudin [1990]. The fact that $K$ is bounded by 1 can be seen with $|K(x, x)| = |q(0)| = \sum_\omega \widehat{q}_\omega = 1$. Finally, note that the Fourier coefficients of an extended k-SoS model can be computed exactly, as in shown *e.g.* later in lemma 1.

## 2.2 Providing certificates with the $F$-norm

As discussed in the previous section our approach for providing a certificate on $f$ relies on first obtaining $\widehat{x}$ using a fast algorithm without guarantees and solving approximately eq. (2) to obtain the certificate (see theorem 1). With this aim, now we need an efficient way to compute the norm $\|\cdot\|_F$. We use here a stochastic approach. Introducing a probability $\widehat{\lambda}_\omega$ (that later will be chosen as a rescaled version of $\widehat{\lambda}_\omega$ in definition 2) on $\mathbb{Z}^d$ we rewrite the $F$-norm

$$\|u\|_F = \sum_{\omega \in \mathbb{Z}^d} \widehat{\lambda}_\omega \cdot \frac{|\widehat{u}_\omega|}{\widehat{\lambda}_\omega} = \mathbb{E}_{\omega \sim \widehat{\lambda}_\omega} \left[ \frac{|\widehat{u}_\omega|}{\widehat{\lambda}_\omega} \right] \tag{7}$$

which yields an objective that is amenable to stochastic optimization. From there, Woodworth et al. [2022] computes a certificate by truncating the sum to a hypercube $\{\omega; \|\omega\|_\infty \leq N\}$ of size

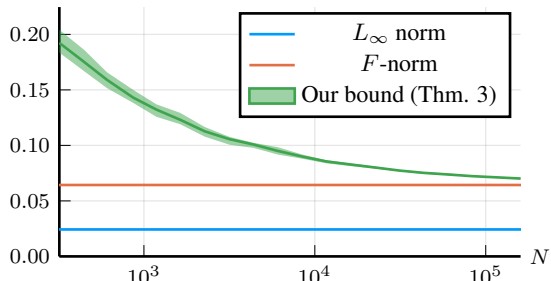

Figure 1: $f - f_\star$ is a trigonometric polynomial approximated by an extended k-SoS model $g$. The $L_\infty$ norm of the difference *(blue)* is upper-bounded by the $F$-norm *(red)*, which is itself upper bounded by the MoM inequality in theorem 3, with probability 98%, here showed with respect to the number $N$ of sampled frequencies. Shaded area shows min/max values across 10 estimations.

$N^d$ and bounding the remaining terms with a smoothness assumption on $u = f - c - g$, which enables to control the decay of $\widehat{u}_\omega$. We want to avoid this cost exponential in the dimension so we proceed differently.

**Probabilistic estimates with the $\mathcal{H}$ norm.** Given that the $F$-norm can be written as an expectation in eq. (7), we approximate it with an empirical mean $\widehat{S}$ given with $N$ i.i.d samples $(\omega_i)_{1 \leq i \leq n} \sim \widehat{\lambda}_\omega$. Now, note that the variance of $\widehat{S}$ can be upper bounded by a Hilbert norm, as

$$\widehat{S} = \frac{1}{N} \sum_{i=1}^{N} \frac{|\widehat{u}_{\omega_i}|}{\lambda_{\omega_i}}, \quad \text{so that} \quad \mathrm{Var}\, \widehat{S} \leq \frac{1}{N} \mathbb{E}\left( \frac{|\widehat{u}_\omega|}{\widehat{\lambda}_\omega} \right)^2 = \frac{1}{N} \sum_{\omega \in \mathbb{Z}^d} \frac{|\widehat{u}_\omega|^2}{\widehat{\lambda}_\omega} = \frac{1}{N} \|u\|_{\mathcal{H}_\lambda}^2, \tag{8}$$

with $\mathcal{H}_\lambda$ the RKHS from definition 2 with kernel $K(x, x') = \sum_{\omega \in \mathbb{Z}^d} \widehat{\lambda}_\omega e^{2\pi \mathrm{i}\omega \cdot (x - x')}$. This allows to quantify the deviation of $\widehat{S}$ from $\mathbb{E}[\widehat{S}] = \|u\|_F$, with *e.g.* Chebychev's inequality, as shown in next theorem.

**Theorem 2** (Certificate with Chebychev Inequality). *Let $(\widehat{\lambda}_\omega)_\omega$ be a probability distribution on $\mathbb{Z}^d$, $\delta \in (0,1)$ and $g$ a positive function. Let $N > 0$ and $\widehat{S}$ be the empirical mean of $|\widehat{f}_\omega - c - \widehat{g}_\omega|/\widehat{\lambda}_\omega$ obtained with $N$ samples $\omega_i \sim \widehat{\lambda}_\omega$. Then, a certificate with probability $1 - \delta$ is given with*

$$f_\star \geq c - \widehat{S} - \frac{\|f - c - g\|_{\mathcal{H}_\lambda}}{\sqrt{N\delta}}, \quad \widehat{S} = \frac{1}{N} \sum_{i=1}^{N} \frac{\left|\widehat{f}_{\omega_i} - c\mathbf{1}_{\omega_i=0} - \widehat{g}_{\omega_i}\right|}{\widehat{\lambda}_{\omega_i}}. \tag{9}$$

*Proof.* From its definition in eq. (7), we see that an unbiased estimator of the $F$-norm is given by $\widehat{S}$. Then, Chebychev's inequality states that $|\widehat{S} - \|u\|_F|^2 \leq \mathsf{Var}\,\widehat{S}/\delta$ with probability at least $1 - \delta$. Using the computation of the variance in eq. (8), it follows that $\|u\|_F \leq \widehat{S} + \|f - c - g\|_{\mathcal{H}_\lambda}/\sqrt{N\delta}$ with probability at least $1 - \delta$. Plugging this expression into eq. (2), we obtain the result. $\square$

Note that the norm in $\mathcal{H}_\lambda$ can be developed with (assuming for conciseness that $(-c)$ is comprised in the 0-frequency of $f$)

$$\|u\|^2_{\mathcal{H}_\lambda} = \sum_{\omega \in \mathbb{Z}^d} \frac{\widehat{f}_\omega^* (\widehat{f}_\omega - 2\widehat{g}_\omega)}{\widehat{\lambda}_\omega} + \|g\|^2_{\mathcal{H}_\lambda} \leq (\|f\|_{\mathcal{H}_\lambda} + \|g\|_{\mathcal{H}_\lambda})^2. \tag{10}$$

Thus, theorem 2 provides a certificate of $f_\star$ as long as we can *(i)* evaluate the Fourier transform $\widehat{g}_\omega$ of $g$ and *(ii)* compute its Hilbert norm in some r.k $\mathcal{H}_\lambda$ induced by $\widehat{\lambda}_\omega$. In next section, we detail the choice we make to achieve this efficiently, with kernels amenable to GPU computation, scaling to thousands of coefficients.

**Remark 2** (Using a RKHS norm instead of the $F$-norm). *Note that since $(\widehat{\lambda}_\omega)_\omega$ sums to $1$, the associated kernel is bounded by $1$. Hence $\|u\|_{L_\infty} \leq \|u\|_{\mathcal{H}_\lambda}$, and the latter could be used instead of the $F$-norm in eq. (2). There are two reasons for taking our approach instead. Firstly, the $F$-norm is always tighter that a RKHS norm (see e.g. [Woodworth et al., 2022, Lem. 4]); secondly, we cannot compute $\|u\|_{\mathcal{H}_\lambda}$ efficiently and have to rely instead on another upper bound. However, taking the number of samples $N = O(\|u\|^2_{\mathcal{H}_\lambda})$ alleviates this issue.*

**Exponential concentration bounds with MoM.** The scaling in $1/\sqrt{\delta}$ in theorem 2 can be prohibitive if one requires a high probability on the result ($\delta \ll 1$). Hopefully, alternative estimator exist for those cases. The Median-of-Mean estimator is an example, illustrated in theorem 3.

**Theorem 3** (Certificate with MoM estimator). *Let $(\widehat{\lambda}_\omega)_\omega$ be a probability distribution on $\mathbb{Z}^d$, and $\delta \in (0,1)$. Draw $N > 0$ frequencies $\omega_i \sim \widehat{\lambda}_\omega$. Define the MoM estimator with the following: for $K \in \mathbb{N}$ s.t. $\delta = e^{-K/8}$ and $N = Kb$, $b \geq 1$, write $B_1, \ldots, B_K$ a partition of $[N]$; then*

$$\mathsf{MoM}_\delta(|\widehat{u}_{\omega_i}|/\lambda_{\omega_i}) = \mathrm{median} \left\{ \frac{1}{b} \sum_{i \in B_j} \frac{|\widehat{f}_{\omega_i} - c\mathbf{1}_{\omega_i=0} - \widehat{g}_{\omega_i}|}{\lambda_{\omega_i}} \right\}_{1 \leq j \leq K}. \tag{11}$$

*A certificate on $f_\star$ with probability $1 - \delta$ follows, with*

$$f_\star \geq c - \mathsf{MoM}_\delta(|\widehat{u}_{\omega_i}|/\lambda_{\omega_i}) - 4\sqrt{2}\,\|f - c - g\|_{\mathcal{H}_\lambda} \sqrt{\frac{\log(1/\delta)}{N}}. \tag{12}$$

*Proof.* Using *e.g.* Theorem 4.1 from Devroye et al. [2016] we get that the deviation of the MoM estimator from the expectation is bounded by

$$\left| \|u\|_F - \mathsf{MoM}_\delta(|\widehat{u}_{\omega_i}|/\lambda_{\omega_i}) \right| \leq 4\sqrt{2} \sqrt{\mathsf{Var}(|\widehat{u}_\omega|/\widehat{\lambda}_\omega) \frac{\log(1/\delta)}{N}} \quad \text{with proba. } 1 - \delta. \tag{13}$$

Using the upper bound on the variance with the $\mathcal{H}_\lambda$ norm given in eq. (8) and plugging the resulting expression into eq. (2), we obtain the result. $\square$

To conclude this section, bounding the $L_\infty$ norm from above with the $F$-norm in eq. (3) enables to obtain a certificate on $f$, as shown in theorem 1. The $F$-norm requires an infinite number of computation in the general case, but can be bounded efficiently with a probabilistic estimate, given by theorem 2 or theorem 3. This is summed up in fig. 1. Note that the difference $\|\cdot\|_F - \|\cdot\|_{L_\infty}$ is a source of conservatism in the certificate which we do not quantify – yet, the $F$-norm is optimal for a class of norms, see [Woodworth et al., 2022, Lemma 3].

## 3 Algorithm and implementation

### 3.1 Bessel kernel

We now detail the specific choice of kernel we make in order to compute the certificate of theorem 2 or theorem 3 efficiently. Our first observation is to use a kernel stable by product, so that we can easily characterize a Hilbert space the model $g$ belongs to. This restricts the choice to the exponential family. That's why we define, for a parameter $s > 0$,

$$\forall x \in \mathbb{T}, \quad q_s(x) = e^{s(\cos(2\pi x)-1)} = \sum_{\omega \in \mathbb{Z}} e^{-s} I_{|\omega|}(s) e^{2\pi i \omega x}, \tag{14}$$

with $I_{|\omega|}(\cdot)$ the modified Bessel function of the first kind [Watson, 1922, p.181]. Then, define $K_s(x, y) = q_s(x - y)$ as in definition 2, and take a tensor product to extend the definition of $K$ to multiple dimension, *i.e.* $K_\mathbf{s}(\mathbf{x}, \mathbf{y}) = \prod_{\ell=1}^d K_{\mathbf{s}_\ell}(\mathbf{x}_\ell, \mathbf{y}_\ell)$ for any $\mathbf{x}, \mathbf{y} \in \mathbb{T}^d$. We refer to this kernel as the *Bessel kernel*, and the associated RKHS as $\mathcal{H}_\mathbf{s}$. It is stable by product as $K_\mathbf{s}(x, y) = K_{\mathbf{s}/2}(x, y)^2$. This is key to compute the Fourier transform of the model $g$, and in contrast to previous approaches which used the exponential kernel with $\widehat{q}_\omega \propto e^{-s|\omega|}$ Woodworth et al. [2022], Bach and Rudi [2023].

In the following, $g$ is a K-SoS model defined as in definition 1, with the Bessel kernel of parameter $\mathbf{s} \in \mathbb{R}_+^d$ defined in eq. (14).

**Lemma 1** (Fourier coefficient of the Bessel kernel)**.** *For $\omega \in \mathbb{Z}^d$, the Fourier coefficient of $g$ in $\omega$ can be computed in $O(drm^2)$ time with*

$$\widehat{g}_\omega = \sum_{i,j=1}^m R_i^\top R_j \prod_{\ell=1}^d e^{-2\mathbf{s}_\ell} I_{|\omega_\ell|}(2s \cos \pi(\mathbf{z}_{i\ell} - \mathbf{z}_{j\ell})) e^{-i\pi\omega_\ell(\mathbf{z}_{i\ell} + \mathbf{z}_{j\ell})}. \tag{15}$$

*Proof.* From its definition in eq. (5), we rewrite $g$ as

$$g(x) = \sum_{i,j=1}^m R_i^\top R_j \prod_{\ell=1}^d K_s(x, \mathbf{z}_{i\ell}) K_s(x, \mathbf{z}_{j\ell}). \tag{16}$$

Now, from the definition of the Bessel kernel in eq. (14), we have that for any $(x, y, z) \in \mathbb{T}$, $K(x, y)K(x, z) = e^{-2s} e^{2s \cos(2\pi(y-z)/2) \cos 2\pi(x-(y+z)/2)}$. By definition of the modified Bessel function, the Fourier coefficient of this expression are given by $I_{|\omega|}(2s \cos(2\pi(y - z)/2))$. Using this into eq. (16), we get the result. $\square$

The second necessary ingredient for using the certificate of theorem 2 is computing a RKHS norm for $g$. It relies on the inclusion of $\mathcal{H}_{2\mathbf{s}}$ into the bigger space of symmetric operator $\mathcal{S}(\mathcal{H}_\mathbf{s})$.

**Lemma 2** (Bound on the RKHS norm of $g$)**.** *$g$ belongs to $\mathcal{H}_{2\mathbf{s}}$, and $\|g\|_{\mathcal{H}_{2\mathbf{s}}}$ is bounded by the Hilbert-Schmidt norm of $\mathcal{S}(\mathcal{H}_\mathbf{s})$, which can be computed in $O(dm^2 + mr^2)$ time, with*

$$\|g\|_{\mathcal{H}_{2\mathbf{s}}}^2 \leq \|g\|_{\mathcal{S}(\mathcal{H}_s)}^2 = \mathrm{Tr}\, (RK_{s,\mathbf{z}}R^\top)^2. \tag{17}$$

*Proof.* Assume that $d = 1$; the reasoning can be extended to multiple dimensions with the tensor product. From the computation of the Fourier coefficient in lemma 1 and the fact that $I_{|\omega|}(2s \cos(2\pi \cdot)) \leq I_{|\omega|}(2s)$, we have that $\widehat{g}_\omega = O(I_{|\omega|}(2s))$ hence $g \in \mathcal{H}_{2s}$. Finally, since the kernel is stable by product, $\mathcal{H}_{2s} = \mathcal{H}_s \odot \mathcal{H}_s$, so we can use *e.g.* [Paulsen and Raghupathi, 2016, Thm. 5.16], with $\mathcal{H}_1 = \mathcal{H}_2 = \mathcal{H}_s$ and $\mathcal{S}(\mathcal{H}_s) = \mathcal{H}_s \otimes \mathcal{H}_s$, with the operator $A = (\varphi(\mathbf{z}_1), \dots, \varphi(\mathbf{z}_m))R^\top R(\varphi(\mathbf{z}_1), \dots, \varphi(\mathbf{z}_m))^* \in \mathcal{S}(\mathcal{H}_s)$. $\square$

With lemma 2, we have that the model $g$ belongs to $\mathcal{H}_{2\mathbf{s}}$, so we will naturally use $\widehat{\lambda}_\omega = \prod_{\ell=1}^{d} e^{-2\mathbf{s}_\ell} I_\omega(2\mathbf{s}_\ell)$ in theorem 2; said differently, the space $\mathcal{H}_\lambda$ introduced in eq. (8) is simply $\mathcal{H}_{2\mathbf{s}}$ defined in eq. (14).

## 3.2 The algorithm: GloptiNets

We can now describe how GloptiNets yields a certificate on $f$. The key observation is that no matter how is obtained our model $g(R, \mathbf{z})$ from definition 1, we will always be able to compute a certificate with theorems 2 and 3. Thus, even though optimizing eq. (9) w.r.t $(c, R, \mathbf{z})$ is highly non-convex, we can use any optimization routine and check empirically its efficiency by looking at the certificate. Finally, thanks to its low-rank structure it is cheaper to evaluate $g$ than evaluating its Fourier coefficient. This is formally shown in proposition 2 in appendix A, where a block-diagonal structure for the model is also introduced. That's why we first optimize $\sup_{c,g} c - \|f - c - g\|_\star$, where $\|\cdot\|_\star$ is a proxy for the $L_\infty$ norm, *e.g.* the log-sum-exp on a random batch of $N$ points[1]:

$$\|f - c - g\|_{L_\infty} \approx \max_{i \in [N]} |f(x_i) - c - g(x_i)| \approx \mathrm{LSE}(f(x_i) - c - g(x_i))_{i \in [N]}. \tag{18}$$

This optimization can be carried out by any deep learning libraries with automatic differentiation and any flavour of gradient ascent. Only afterwards do we compute the certificate with theorems 2 and 3. This procedure is summed up in alg. 1.

---

**Algorithm 1:** GloptiNets

---

**Data:** A trigonometric polynomial $f$, a candidate $z$ s.t. $c = f(z)$, a model $g$, and a probability $\delta$.
**Result:** A certificate $|f_\star - f(z)| \leq \epsilon_\delta$ with proba. $1 - \delta$.
```
/* Optimize g with function values                                        */
```
**for** *epoch = 1:nepochs* **do**
    Sample $x_1, \ldots, x_N \in \mathbb{T}^d$ ;
    $L, \nabla L = \mathrm{autodiff}(\mathrm{LSE}(f(x_i) - c - g(x_i))_{i \in [N]})$ ;
    $\mathbf{z}, R \leftarrow \mathrm{optimizer}(\nabla L)$ ;
```
/* Compute a certificate                                                  */
```
$\widehat{\lambda}_\omega$: probability distribution on $\mathbb{Z}^d$ with $\widehat{\lambda}_\omega = \prod_{\ell=1}^{d} e^{-2\mathbf{s}_\ell} I_\omega(2\mathbf{s}_\ell)$;
Sample $\Omega = (\omega_1, \ldots, \omega_N) \sim \widehat{\lambda}_\omega$ ;
Compute $M = \mathrm{MoM}_\delta(|\widehat{f}_{\omega_i} - c\mathbf{1}_{\omega_i=0} - \widehat{g}_{\omega_i}|/\lambda_{\omega_i})_{i \in [n]}$ and $\bar{\sigma} = \|g\|_{\mathcal{S}(\mathcal{H}_\mathbf{s})}$;
Returns $\epsilon_\delta = c - M - 4\sqrt{2}\bar{\sigma}\sqrt{\log(1/\delta)/N}$;

---

**Remark 3** (Providing a candidate). *In alg. 1, a candidate estimate $c$ for the minimum value $f(x_\star)$ is necessary. However, it is possible to overcome this requirement by incorporating $c$ as a learnable parameter within the training loop. Moreover, $x_\star$ can be learned using techniques similar to those in Rudi et al. [2020]: by replacing the lower bound $c$ with a parabola centered at $z$, $z$ becomes a candidate for $x_\star$ with precision corresponding to the tightness of the certificate. Note however that this method introduces additional hyperparameters.*

## 3.3 Specific implementation for the Chebychev basis

As already observed in Bach and Rudi [2023], a result on trigonometric polynomial on $\mathbb{T}^d$ directly extends to a real polynomials on $[-1, 1]^d$. The reason for that is that minimizing $h$ on $[-1, 1]^d$ amounts to minimizing the trigonometric polynomial $f = h((\cos 2\pi x_1, \ldots, \cos 2\pi x_d))$ on $\mathbb{T}^d$. Note however that $f$ is an even function in all dimension, as for any $x \in \mathbb{T}^d$, $f(x) = f(x_1, \ldots, -x_i, \ldots, x_d)$. Thus, approximating $f - f_\star$ with a K-SoS model of definition 1 is suboptimal, in the sense that we could approximate $f$ only on $[0, 1/2]^d$, which is $2^{-d}$ smaller. Put differently, the Fourier coefficient of $f$ are real by design: it would be convenient to enforce this structure in the model $g$. This is achieved with proposition 1.

---

[1] Another detail of practical importance is that this loss can be efficiently backpropagated through; on the other hand, the certificate is not easily vectorized, and the Bessel function involved would require specific approximation to be efficiently backpropagated through.

**Proposition 1** (Kernel defined on the Chebychev basis). *Let $q$ be a real, even function on the torus, bounded by 1, as in eq. (6). Let $K$ be the kernel defined on $[-1, 1]$ by*

$$\forall (u, v) \in (0, ^1/_2), K(\cos 2\pi u, \cos 2\pi v) = \frac{1}{2}(q(u + v) + q(u - v)). \tag{19}$$

*Then $K$ is a symmetric, p.d., hence reproducing kernel, bounded by 1, with explicit feature map given by*

$$\forall (x, y) \in [-1, 1], K(x, y) = \widehat{q}_0 + \sum_{\omega \in \mathbb{N}} 2\widehat{q}_\omega H_\omega(x) H_\omega(y). \tag{20}$$

The proof is available in appendix B. It simply relies on expanding the definition of $K$ in eq. (19). The resulting expression in eq. (20) exhibits only cosine terms (in the decomposition of $x \mapsto K(\cos 2\pi x, y)$). This enables to directly extend the PSD models from definition 1 with such kernels. Finally, when used with the Bessel kernel of eq. (14), we recover an easy computation of the Chebychev coefficient, as shown in lemma 3, in $O(drm^2)$ time. This enables to approximate any function expressed on the Chebychev basis. Note that polynomials expressed in other basis can be certified too, by first operating a change of basis.

## 4 Experiments

The code to reproduce these experiments is available at

`github.com/gaspardbb/GloptiNets.jl`

**Settings.** Given a function $h$, we compute a candidate $\widehat{x}$ with gradient descent and multiple initializations. The goal is then to certify that $\widehat{x}$ is indeed a global minimizer of $h$. This is a common setup in the Polynomial-SoS literature Wang and Magron [2022]. To illustrate the influence of the number of parameters, the positive model $g$ defined in definition 1 for GloptiNets designates either a small model GN-small with 1792 parameters, or a bigger model GN-big with 22528 parameters. The latter should have higher expressivity and better interpolate positive functions, leading to tighter certificates. All results for GloptiNets are obtained with confidence $1 - \delta = 1 - e^{-4} \geq 98\%$. All other details regarding the experiments are reported in appendix C.

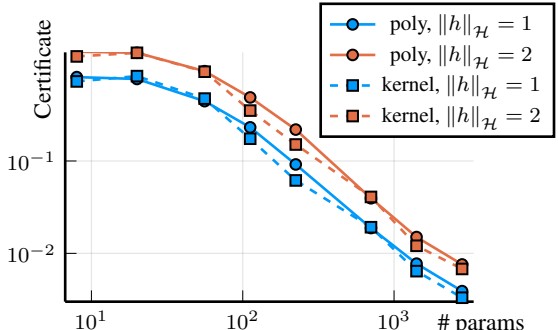

Figure 2: Certificate *vs.* number of parameters in $g$, for a given function $h$. The higher the RKHS norm of $h$, the more difficult it is to approximate uniformly and the looser the certificate, independently of the function type. The more parameters in the k-SoS model, the tighter the certificates obtained with theorem 3.

**Polynomials.** We first consider the case where $h$ is a random trigonometric polynomial. Note that this is a restrictive analysis, as GloptiNets can handle any smooth functions (*i.e.* with infinite non-zero Fourier coefficients). Polynomials have various dimension $d$, degree $p$, number of coefficients $n$, but a constant RKHS norm $\mathcal{H}_{2\mathbf{1}_d}$. We compare the performances of GloptiNets to TSSOS, in its complex polynomial variant Wang and Magron [2022]. The latter is used with parameters such that it executes the fastest, but without guarantees of convergence to the global minimum $f_\star$. Table 1 shows the certificates $h(x_\star) - h(\widehat{x})$ and the execution times (lower is better, $t$ in seconds) for TSSOS, GN-small and GN-big. Figure 2 provides certificate on a random polynomial, function of the number of parameters in $g$.

**Kernel mixtures.** While polynomials provide ground for comparison with existing work, GloptiNets is not confined to this function class. This is evidenced by experiments on kernel mixtures, where our approach stands as the only viable alternative we are aware of. The function we certify are of the form $h(x) = \sum_{i=1}^n \alpha_i K(x_i, x)$, where $K$ is the Bessel kernel of eq. (14). Kernel mixtures are ubiquitous in machine learning and arise *e.g.* when performing kernel ridge regression. Certificates obtained on mixtures are compared with those obtained on polynomials in fig. 2, function of the model size $g$.

Table 1: GloptiNets and TSSOS on random trigonometric polynomials. While TSSOS provides machine-precision certificates, its running time grows exponentially with the problem size, and eventually fails on problems 3 and 6. On the other hand, GloptiNets has constant running time no matter the problem size, and its certificates can be tightened by increasing the model size.

| $d$ | $p$ | $n$ | TSSOS | | GN-small | | GN-big | |
|---|---|---|---|---|---|---|---|---|
| | | | Certif. | $t$ | Certif. | $t$ | Certif. | $t$ |
| | 5 | 85 | $5.3 \cdot 10^{-11}$ | 3 | $8.35 \cdot 10^{-4}$ | $6 \cdot 10^3$ | $2.64 \cdot 10^{-4}$ | $9 \cdot 10^3$ |
| 3 | 7 | 231 | $4.7 \cdot 10^{-13}$ | 120 | $9.51 \cdot 10^{-4}$ | $6 \cdot 10^3$ | $2.90 \cdot 10^{-4}$ | $9 \cdot 10^3$ |
| | 9 | 489 | out of memory! | - | $1.18 \cdot 10^{-3}$ | $6 \cdot 10^3$ | $3.34 \cdot 10^{-4}$ | $9 \cdot 10^3$ |
| | 3 | 33 | $3.1 \cdot 10^{-10}$ | 0.1 | $2.46 \cdot 10^{-2}$ | $1 \cdot 10^4$ | $3.45 \cdot 10^{-3}$ | $2 \cdot 10^4$ |
| 4 | 5 | 225 | $4.8 \cdot 10^{-12}$ | 53 | $3.71 \cdot 10^{-2}$ | $1 \cdot 10^4$ | $3.59 \cdot 10^{-3}$ | $2 \cdot 10^4$ |
| | 7 | 833 | out of memory! | - | $4.76 \cdot 10^{-2}$ | $1 \cdot 10^4$ | $4.85 \cdot 10^{-3}$ | $2 \cdot 10^4$ |

**Results.** There are two key hindsight about the performances of GloptiNets. Firstly, its certificate *does not depend on the structure* of the function to optimize. Thus, although GloptiNets does not match the performances of TSSOS on small polynomials, it can tackle polynomials which cannot be handled by competitors, with arbitrarily as many coefficients ($n = \infty$). For instance, TSSOS cannot handle problems with $n \in \{489, 833\}$ in table 1. More importantly, GloptiNets can certify a richer class of functions than polynomials, among which kernel mixtures. The performances of GloptiNets mostly depends on the complexity of the function to certify, as measured with its RKHS norm.

Secondly, note that *a bigger model yields tighter certificate*. This is detailed in fig. 2, where the same function $f$ is optimized with various models. The dependency of the certificate on the norm of $f$ is shown in fig. 3 in appendix C, along with experiments with Chebychev polynomials.

## 5   Limitations

One limitation of GloptiNets is the trade-off resulting from its high flexibility for obtaining a certificate as in alg. 1. While this flexibility offers numerous advantages, it also introduces the need for an extensive hyperparameter search. Although we have identified a set of hyperparameters that align with deep learning practices – utilizing a Momentum optimizer with cosine decay and a large initial learning rate – the optimal settings may vary depending on the specific problem at hand.

In the same vein, the certificates given by GloptiNets are of moderate accuracy. While adding more parameters into the k-SoS model certainly helps (as shown in fig. 2), alternative optimization scheme to interpolate $h - h(\hat{x})$ with $g$ might provide easier improvement. For instance, we found that using approximate second-order scheme in alg. 1 is key to obtaining good certificates.

In the specific settings of polynomial optimization, we highlight that our model is not competitive on problems which exhibits some algebraic structure, as for instance term sparsity or the constant trace property. Typically, problems with coefficients of low degrees (less or equal than 2), which encompass notably the OPF problem, are really well handled by the family of solvers TSSOS belongs to. Finally, GloptiNets does not handle constraints yet.

## 6   Conclusion

The GloptiNets algorithm presented in this work lays the foundation for a new family of solvers which provide certificates to non-convex problems. While our approach does not aim to replace the well-established Lasserre's hierarchy for sparse polynomials, it offers a fresh perspective on tackling a new set of problems at scale. Through demonstrations on synthetic examples, we have showcased the potential of our approach. Further research directions include extensive parameter tuning to obtain tighter certificates, with the possibility of leveraging second-order optimization schemes, along with warm-restart schemes for application which requires solving multiple similar problems sequentially.

**Acknowledgments.** AR acknowleges support of the French government under management of Agence Nationale de la Recherche as part of the "Investissements d'avenir" program, reference ANR-19-P3IA-0001 (PRAIRIE 3IA Institute). AR acknowledges support of the European Research Council (grant REAL 947908). JM was supported by the ERC grant number 101087696 (APHE-LAIA project) and by ANR 3IA MIAI@Grenoble Alpe (ANR-19-P3IA-0003).

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

# A  Extensions

We explore additional extensions of GloptiNets that further enhance its appeal. We first describe a block diagonal structure for the model for faster evaluation, a theoretical splitting scheme for optimization, and finally a warm-start scheme.

## A.1  Block diagonal structure for efficient computation

Without any further assumption, we see that a model from definition 1 can be evaluated in $O(drm)$ time; its Fourier coefficient given by lemma 1 in $O(dm^2r)$; the bound on the RKHS norm is computed in $O(dm^2 + mr^2)$ time thanks to lemma 2; all that enables to compute a certificate, as stated in theorem 2, in $O(Ndm^2r + mr^2)$ time, where $N$ is the number of frequencies sampled. If the function $f$ to be minimized has big $\mathcal{H}_{\mathbf{s}}$ norm, we might need a large model size $m$ to have $f - f_\star \approx g$. Hence, we introduce specific structure on $G$ which makes it *block-diagonal* and *better conditioned*.

**Proposition 2** (Block-diagonal PSD model)**.** *Let $g$ be a PSD model as in definition 1, with $m = bs$ anchors. Split them into $b$ groups, denoting them $\mathbf{z}_{ij}$, $i \in [b]$ and $j \in [s]$. Compute the Cholesky factorization of each kernel matrix $T_i^\top T_i = K_{\mathbf{z}_i} \in \mathbb{R}^{s \times s}$. Then, define $G$ as a block-diagonal matrix, with $b$ blocks defined as $G_i = \tilde{R}_i \tilde{R}_i^\top$, $\tilde{R}_i = T^{-1} R_i$, and $R_i \in \mathbb{R}^{r \times s}$. Equivalently,*

$$
G = \begin{pmatrix} \tilde{R}_1 \tilde{R}_1^\top & & \\ & \ddots & \\ & & \tilde{R}_b \tilde{R}_b^\top \end{pmatrix}, \quad s.t. \quad g(x) = \sum_{i=1}^{b} \left\| \tilde{R}_i^\top K_{\mathbf{z_i}}(x) \right\|^2, \quad K_{\mathbf{z_i}}(x) = K(\mathbf{z}_{ij}, x)_{1 \leq j \leq s}.
$$
(21)

*Then $g$ can be evaluated in $O(rbs^3 d)$ time, $\widehat{g}_\omega$ in $O(bs^2(dr + s))$ time, and $\|g\|^2_{\mathcal{S}(\mathcal{H}_s)}$ in $O(b^2(rs^2 + r^2s) + bs^3)$ time. The model has $(r + d)bs$ real parameters.*

*Proof.* Having $G$ defined as such, it is psd, of rank at most $rb \leq sb = m$. Written $g(x) = \sum_{i=1}^{b} \|\tilde{R}_i^\top K_{\mathbf{z_i}}(x)\|^2$, we can compute the Fourier coefficient by applying lemma 1 to each of the $b$ component. Adding the cost of computing $G_i = \tilde{R}_i \tilde{R}_i^\top$ results in complexity of $O(bs^2(dr + s))$. Finally, note that $\|g\|^2_{\mathcal{S}(\mathcal{H}_s)} = \|A\|^2_{\mathcal{S}(\mathcal{H}_s)}$ where

$$
A = ((\varphi(\mathbf{z}_{1j}))_{j \in [s]}, \ldots, (\varphi(\mathbf{z}_{bj}))_{j \in [s]})(\operatorname{Diag} G_i)_{i \in [b]}((\varphi(\mathbf{z}_{1j}))_{j \in [s]}, \ldots, (\varphi(\mathbf{z}_{bj}))_{j \in [s]})^*.
$$

Then, defining $Q$ the matrix of $b \times b$ blocks of size $s \times s$ s.t. for $j, k \in [b]$, $Q_{jk} = K(\mathbf{z}_j, \mathbf{z}_k) \in \mathbb{R}^{s \times s}$, we have

$$
\|A\|^2_{\mathcal{S}(\mathcal{H}_s)} = \operatorname{Tr} Q(\operatorname{Diag} G_i)_{i \in [b]} Q(\operatorname{Diag} G_i)_{i \in [b]} = \sum_{j,k=1}^{b} \operatorname{Tr} G_j Q_{jk} G_k Q_{kj},
$$
(22)

and each term in the sum can be written $\operatorname{Tr}(\tilde{R}_j^\top Q_{jk} \tilde{R}_k)(\tilde{R}_k^\top Q_{kj} \tilde{R}_j^\top) = \|\tilde{R}_j^\top Q_{jk} \tilde{R}_k\|^2_{HS}$, which is computed in $O(rs^2 + r^2s)$ time, plus $O(bs^3)$ to compute the Cholesky factor. $\square$

Denoting $\varphi_{\mathbf{z}_i} = (\varphi(\mathbf{z}_{ij}))_{1 \leq j \leq s}$, note that

$$
\varphi_{\mathbf{z}_i} G_i \varphi_{\mathbf{z}_i}^* = \varphi_{\mathbf{z}_i} T_i^{-1} R_i R_i^\top (\varphi_{\mathbf{z}_i} T_i^{-1})^* = E_i R_i R_i^\top E_i^*,
$$
(23)

with $E_i = \varphi_{\mathbf{z}_i} T_i^{-1}$ an orthonormal basis of $\operatorname{Span}(\varphi_{\mathbf{z}_{ij}})_{1 \leq j \leq s}$ as $E_i^* E_i = \mathbf{I}_s$. Thus, each model's coefficient is defined on an orthonormal basis, which makes the optimization easier. Of course, this comes at an added $s^3$ complexity, which could be alleviated by using *e.g.* an incomplete Cholesky factorization instead.

**Remark 4** (Relation to Term Sparsity in POP)**.** *The successful application of polynomial hierarchies to problems with thousands of variables rely on making the moment matrix $M$ having a block structure Wang et al. [2021b,a]. If the monomial basis has size $m$, the constraint $M \succeq 0$ is replaced with $M = (\operatorname{Diag} M_i)_{i \in [b]}$ and $M_i \succeq 0$. This enables to solve $b$ SDP of size at most $s$ instead of one of size $m$. Our model in proposition 2 follows a similar route for having a lower computational budget.*

## A.2 Global optimization with splitting scheme

While GloptiNets can provide certificates for functions, it falls behind local solvers in terms of competitiveness. The challenge lies in the fact that finding a certificate is considerably more difficult than finding a local minimum, as it necessitates the uniform approximation of the entire function. However, we present a novel algorithmic framework that has the potential to enhance the competitiveness of GloptiNets with local solvers while simultaneously delivering certificates. Our approach involves partitioning the search domain into multiple regions and computing lower bounds for each partition. By discarding portions of the domain where we can certify that the function exceeds a certain threshold, the algorithm progressively simplifies the optimization problem and removes areas from consideration. Moreover, such an approach is naturally well suited to parallel computation.

The algorithm relies on a divide-and-conquer mechanism. First, we split the hypercube $(-1, 1)^d$ in $N$ regions, where $N$ is the number of core available. We compute an upper bound with a local solver. For each region, we run GloptiNets *in parallel*, computing a certificate at regular interval. As soon as the certificate is bigger than the upper bound, we stop the process: we know that the global minimum is not in the associated region. We can then reallocate the freed computing power by splitting the biggest current region, which yields an easier problem. We stop as soon as the region considered are small enough. This is summarized in alg. 2, where $\textcircled{P}$ indicates the loop run in parallel.

Note that minimizing $f$ on a hypercube of center $\mu$ and size $\sigma$ amounts to minimizing $x \mapsto f((x - \mu)/\sigma)$ on $[-1, 1]^d$, which is another Chebychev polynomial whose coefficients can be evaluated efficiently thanks to the order-2 relation every orthonormal polynomial satisfy. For Chebychev polynomials, that is $H_{\omega+1}(x) = 2xH_\omega(x) - H_{\omega-1}(x)$.

---

**Algorithm 2:** Splitting scheme with GloptiNets

---

**Data:** A Chebychev polynomial $f$ with a unique global optimum, a probability $\delta$, a number of cores $N$ and a volume $\rho < 1/N$.

**Result:** A certificate on $f$: $f_\star \geq C_\delta(f)$ with proba. $1 - \delta_\star$.

```
/* Initialization:  upper bound and partition                       */
```
$\Pi = \mathsf{partition}([-1, 1]^d, N), \delta_\star = 0$ ;

$\textcircled{P}$ $\mathsf{ub} = \min_{\pi \in \Pi} \{\mathsf{localsolver}_{x \in \pi} f(x)\}$;

```
/* Iterate over the partition                                       */
```
$\textcircled{P}$ **for** $\pi \in \Pi$, **While** $\mathsf{length}(\Pi) > 1$ **do**

    **while** $C_\delta(f_\pi) < \mathsf{ub}$ **do**
        $\lfloor$ Continue optimization;

    Split biggest part: $\pi_0 = \arg\max_{\pi \in \Pi} \mathrm{Vol}(\pi)$; $(\pi_1, \pi_2) = \mathsf{partition}(\pi_0, 2)$ ;

    If $\mathrm{Vol}(\pi_{1,2}) < \rho$: end this process ;

    Update upper bound: $\mathsf{ub} = \min\{\mathsf{ub}, \mathsf{localsolver}_{x \in \pi_{1,2}} f(x)\}$ ;

    Update search space and $\delta_\star$: $\Pi = \Pi \setminus \{\pi, \pi_0\} \cup \{\pi_1, \pi_2\}, \delta_\star = 1 - (1 - \delta_\star)(1 - \delta)$;

```
/* A single region in Π remains                                     */
```
Returns $\Pi = \{\pi\}, C_\delta(f_\pi), \delta_\star$;

---

## A.3 Warm restarts

Our model distinguishes itself by leveraging the analytical properties of the objective function, rather than relying solely on algebraic characteristics. This approach offers a notable advantage, as closely related functions can naturally benefit from a warm restart. For example, if we already have a certificate for a function $f$ using a PSD model $g$, and we seek to compute a certificate for a similar function $\tilde{f} \approx f$, we can readily employ GloptiNets by initializing the PSD model with $g$. Indeed, if $f - f_\star \approx g$, we can expect $\tilde{f} - \tilde{f}_\star \approx g$, so we can expect the optimization to be faster.

In contrast, P-SoS methods, which rely on SDP programs, cannot directly adapt to new problems without significant effort. For instance, if a new component is introduced, an entirely new SDP must be solved. Our model's ability to accommodate related yet distinct problems could prove highly valuable in domains with a frequent need to certify different but closely related problems. In the industry, the Optimal Power Flow (OPF) problem requires periodic solves every 5 minutes

Van Hentenryck [18]. With GloptiNets, once the initial challenging solve is performed, subsequent solves become easier assuming minimal changes in supply and demand conditions.

## A.4 Optimizing the certificate directly

As explained in section 3.2 where GloptiNets is introduced, we optimize a proxy of the $L_\infty$ norm rather than the certificate of theorems 2 and 3. This proxy is the log-sum-exp on a random batch of $N$ points. The reason for this is that evaluating an extended k-SoS model $g(x)$ on $x \in \mathbb{T}^d$ requires $O(drs)$ time, while evaluating $\widehat{g}_\omega$ on $\omega \in \mathbb{Z}^d$ requires $O(drs^2)$ time. Yet, optimizing the certificate directly could probably help obtaining higher-precision certificate. Lemma 4 in appendix D sketches a method to reduce the computational cost of the Fourier computation from $O(s^2)$ to $O(s)$.

## B Kernel defined on the Chebychev basis

In this section we describe the approach we take to model functions written in the Chebychev basis. For $h$ such a polynomial, a naive approach would simply model $f = h \circ \cos(2\pi\cdot)$ as a trigonometric polynomial. However, note that the decomposition of $f$ only has cosine terms. Thus, approximating $f - f_\star$ efficiently requires a PSD model which has only cosine terms in its Fourier decomposition. This is achieved by using a kernel written in the Chebychev basis, as introduce in proposition 1, for which we now provide a proof.

*Proof of proposition 1.* Let $x, y \in [-1, 1]$ and $u, v \in [0, 1/2]$ s.t. $x, y = \cos(2\pi u), \cos(2\pi v)$, by bijectivity of the cosine function on $[0, \pi]$. From the definition of $K$ in eq. (19) and the definition of $q$ in eq. (6), we have that

$$
\begin{aligned}
K(x, y) &= \frac{1}{2} \sum_{\omega \in \mathbb{Z}} \widehat{q}_\omega \left( e^{2\pi i \omega(u+v)} + e^{2\pi i \omega(u-v)} \right) \\
&= \sum_{\omega \in \mathbb{Z}} \widehat{q}_\omega e^{2\pi i \omega u} \cos(2\pi\omega v) \\
&= \widehat{q}_0 + 2 \sum_{\omega \in \mathbb{N}} \widehat{q}_\omega \cos(2\pi\omega u) \cos(2\pi\omega v) \\
&= \widehat{q}_0 + 2 \sum_{\omega \in \mathbb{N}} \widehat{q}_\omega H_\omega(u) H_\omega(v).
\end{aligned}
$$

Since $q$ has positive Fourier transform, this makes the feature map of $K$ explicit with $K(x, y) = \varphi(u) \cdot \varphi(v)$, $\varphi(u)_\omega = \sqrt{(1 + \mathbf{1}_{\omega \neq 0})\widehat{q}_\omega} H_\omega(u)$, for $\omega \in \mathbb{N}$. Hence the kernel is a reproducing kernel. $\qquad\square$

We now use this kernel with the Bessel function $x \mapsto e^{s(\cos(2\pi x)-1)}$, *i.e.* we define the kernel $K$ on $[-1, 1]$ to satisfy

$$
\forall u, v \in (0, 1/2), \quad K(\cos(2\pi u), \cos(2\pi v)) = \frac{1}{2} \left( e^{s(\cos(2\pi(u+v)))} + e^{s(\cos(2\pi(u-v)))} \right). \tag{24}
$$

As it was the case for the torus, this kernel enables an easy characterization of a RKHS in which an associated PSD model $g$ lives.

**Lemma 3** (Chebychev coefficient of the Bessel kernel). *Let $g$ be a PSD model as in definition 1, with the kernel $K$ of eq. (24). Then, the Chebychev coefficient $\omega \in \mathbb{N}^d$ of $g$ can be computed in $O(rdm^2)$ time with*

$$
g_\omega = \sum_{i,j=1}^m R_i^\top R_j \prod_{\ell=1}^d (1 + \mathbf{1}_{\omega \neq 0}) \frac{e^{-2\mathbf{s}_\ell}}{2} \Bigg[ I_{\omega_\ell}(2\mathbf{s}_\ell \sigma_{-\ell ij}) H_{\omega_\ell}(\sigma_{+\ell ij}) \tag{25}
$$
$$
+ I_{\omega_\ell}(2\mathbf{s}_\ell \sigma_{+\ell ij}) H_{\omega_\ell}(\sigma_{-\ell ij}) \Bigg]
$$

*where*

$$
\sigma_{\pm \ell ij} = \cos(2\pi m_{\pm \ell ij}), \quad m_{\pm \ell ij} = (\mathbf{u}_{\ell ij} \pm \mathbf{u}_{\ell ij})/2, \quad \text{and} \quad \cos 2\pi\mathbf{u}_{\ell ij} = \mathbf{z}_{\ell ij}.
$$

*Proof.*

**Expanding $g$ and definition of Chebychev coefficient.** From the definition of $g$ in eq. (5), we have

$$g(\mathbf{x}) = \sum_{i,j=1}^{m} R_i^\top R_j \prod_{\ell=1}^{d} K_{\mathbf{s}_\ell}(\mathbf{x}_\ell, \mathbf{z}_{\ell i}) K_{\mathbf{s}_\ell}(\mathbf{x}_\ell, \mathbf{z}_{\ell j}). \tag{26}$$

We consider $x, y, z \in (-1, 1)$ and $s > 0$. We denote $u, v, w \in (0, 1/2)$ s.t.

$$x, y, z = \cos 2\pi u, \cos 2\pi v, \cos 2\pi w$$

with the bijectivity of $x \mapsto \cos(2\pi x)$ on $(0, 1/2)$. We now compute the Chebychev coefficient of $x \mapsto K_s(x, y) K_s(x, z)$. Denoted $p_\omega$, this is

$$\forall \omega \in \mathbb{N}, \ p_\omega = \frac{1 + \mathbf{1}_{\omega \neq 0}}{\pi} \int_{-1}^{1} K_s(x, y) K_s(x, z) T_\omega(x) \frac{\mathrm{d}x}{\sqrt{1 - x^2}},$$

or equivalently

$$\forall \omega \in \mathbb{N}, \ p_\omega = (1 + \mathbf{1}_{\omega \neq 0}) \int_{0}^{1} K_s(\cos 2\pi u, \cos 2\pi v) K_s(\cos 2\pi u, \cos 2\pi w) \cos(2\pi\omega u) \mathrm{d}u. \tag{27}$$

**Chebychev coefficient of kernel product.** With the definition of the kernel in proposition 1, eq. (19), we have

$$K_s(x, y) K_s(x, z) = \frac{1}{4} \left( p(u + v) + p(u - v) \right) \times \left( p(u + w) + p(u - w) \right)$$

$$= \frac{e^{-2s}}{4} \left( e^{s \cos 2\pi(u+v)} + e^{s \cos 2\pi(u-v)} \right) \times \left( e^{s \cos 2\pi(u+w)} + e^{s \cos 2\pi(u-w)} \right)$$

Now use the sum-to-product formula with the cosines to obtain

$$K_s(x, y) K_s(x, z) = \frac{e^{-2s}}{4} \left( e^{2s \cos 2\pi(\frac{v-w}{2}) \cos 2\pi(u + \frac{v+w}{2})} + e^{2s \cos 2\pi(\frac{v-w}{2}) \cos 2\pi(u - \frac{v+w}{2})} \right.$$
$$\left. + e^{2s \cos 2\pi(\frac{v+w}{2}) \cos 2\pi(u + \frac{v-w}{2})} + e^{2s \cos 2\pi(\frac{v+w}{2}) \cos 2\pi(u - \frac{v-w}{2})} \right), \tag{28}$$

We simplify this expression by introducing

$$m_\pm = \frac{1}{2}(v \pm w) \ \text{ and } \ \sigma_\pm = \cos 2\pi m_\pm. \tag{29}$$

Then, eq. (28) becomes

$$K_s(x, y) K_s(x, z) = \frac{e^{-2s}}{4} \left( e^{2s\sigma_- \cos 2\pi(u+m_+)} + e^{2s\sigma_- \cos 2\pi(u-m_+)} \right. \tag{30}$$
$$\left. + e^{2s\sigma_+ \cos 2\pi(u+m_-)} + e^{2s\sigma_+ \cos 2\pi(u-m_-)} \right).$$

We recognize the definition of the kernel (which is not a surprise as we chose the kernel to be stable by product). However, we need variables in $(0, 1/2)$ to retrieve the proper definition of the kernel. Instead, we use lemma 5 on eq. (30) combined with eq. (27), to obtain

$$p_\omega = (1 + \mathbf{1}_{\omega \neq 0}) \frac{e^{-2s}}{4} \left( \cos(2\pi\omega m_+) I_\omega(2s\sigma_-) + \cos(2\pi\omega m_+) I_\omega(2s\sigma_-) \right.$$
$$\left. + \cos(2\pi\omega m_-) I_\omega(2s\sigma_+) + \cos(2\pi\omega m_-) I_\omega(2s\sigma_+) \right),$$

which gives

$$p_\omega = (1 + \mathbf{1}_{\omega \neq 0}) \frac{e^{-2s}}{2} (\cos(2\pi\omega m_+) I_\omega(2s\sigma_-) + \cos(2\pi\omega m_-) I_\omega(2s\sigma_+)). \tag{31}$$

Equation (31) contains the Chebychev coefficient of the product of two kernel function as defined in eq. (27). Plugging this result into the definition of $g$ in eq. (26), and noting that $\cos(2\pi\omega m_\pm) = H_\omega(\cos 2\pi m_\pm) = H_\omega(\sigma_\pm)$, we obtain the result. $\qquad \square$

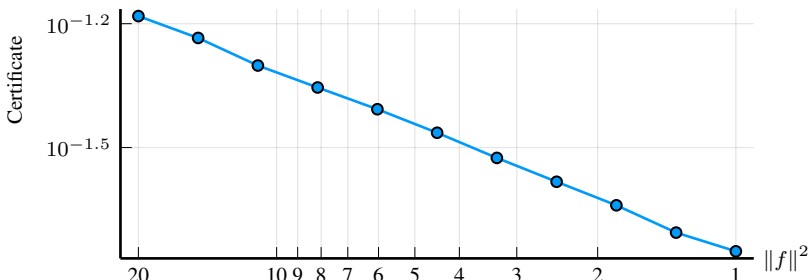

Figure 3: Certificate *vs.* RKHS norm of $f$, for a given model $g$ with a fixed number of parameters. $f$ has 1146 coefficients and $g$ has 22528 parameters. Best certificate is kept among a set of optimization hyperparameters. As the norm of $f$ decreases, fitting $f - f_\star$ with $g$ is easier and the certificate becomes tighter.

Thanks to lemma 3, we see that a model $g$ defined as in definition 1 with the Bessel kernel $K_{\mathbf{s}}$ of eq. (24) as its Chebychev coefficients decaying in $O(I_\omega(2s))$. Hence, it belongs to $\mathcal{H}_{2\mathbf{s}}$, the RKHS associated to $K_{2\mathbf{s}}$.

## C  Additional details on the experiments

**Tuning the hyperparameters.** The time reported in section 4 does *not* take into account the experiments needed to find a good set of hyperparameters. The parameters tuned were the type of optimizer, the decay of learning rate, and the regularization on the Frobenius norm of $G$.

**Regularization.** Regularization is performed by approximating the $HS$ norm with a proxy which is faster to compute. We use $\|R_j^\top R_k\|_{HS}^2$ instead of $\|\tilde{R}_j^\top Q_{jk} \tilde{R}_k\|_{HS}^2$ in eq. (22).

**Hardware.** GloptiNets was used with NVIDIA V100 GPUs for the interpolation part, and Intel Xeon CPU E5-2698 v4 @ 2.20GHz for computing the certificate. TSSOS was run on a Apple M1 chip with Mosek solver.

**Configuration of TSSOS.** We use the lowest possible relaxation order $d$ (*i.e.* $\lceil \deg f/2 \rceil$), along with Chordal sparsity. We use the first relaxation step of the hierarchy. In these settings, TSSOS is not guaranteed to converge to $f_\star$ but will executes the fastest.

**Certificate vs. number of parameter for a given function.** In fig. 2, the target function is a random polynomial of norm 1 or 2, or a kernel mixture with 10 coefficients of norm 1 or 2. The models forming the blue line are defined as in proposition 2, with rank, block size and number of blocks equal to $(1, bs, 1)$ respectively, with $bs$ the block size we vary. The number of frequencies sampled to compute the certificate is $1.6 \cdot 10^7$, and accounts for the fact that the bound on the variance becomes larger than the MOM estimator for large models.

**Certificate vs. problem difficulty for a given model.** We have 3 related parameters: the quality of the optimization (given by the certificate), the expressivity of the model (given by its number of parameters), and the difficulty of the optimization (given by the norm of the function). In fig. 3, we fix the latter and plot the relation between the first two. Here, we fix the model with parameters $(8, 16, 128)$, and we optimize a polynomial in $3d$ of degree 12, with RKHS norm ranging from 1 to 20. The certificates obtained are given in fig. 3. The resulting plot exhibits a clear polynomial relation between the certificate and the norm of the function, with a slope of $-0.88$. This suggest that the certificate behaves as $O(\|f\|_{\mathcal{H}_{2\mathbf{s}}}^{1/2})$.

**Comparison with TSSOS on the Fourier basis.** In table 1, the polynomials $f$ all have a RKHS norm of 1. The small model is defined as in proposition 2, with rank, block size and number of blocks equal to $4, 32, 8$ respectively. For the big models, those values are $8, 128, 16$. The certificate is the

Table 2: GloptiNets and TSSOS on random Chebychev polynomials. The same conclusion as in table 1 applies. While TSSOS is very efficient on small problems, its memory requirements grow exponentially with the problem size. GloptiNets has less accuracy, but a computational burden which does not increase with the problem size.

| $d$ | $p$ | $n$ | TSSOS | | GN-small | | GN-big | |
|---|---|---|---|---|---|---|---|---|
| | | | Certif. | $t$ | Certif. | $t$ | Certif. | $t$ |
| | 3 | 255 | $3.4 \cdot 10^{-7}$ | 6 | $1.1 \cdot 10^{-2}$ | $2 \cdot 10^2$ | $4.1 \cdot 10^{-3}$ | $1 \cdot 10^3$ |
| 4 | 4 | 624 | $2.1 \cdot 10^{-9}$ | 153 | $2.5 \cdot 10^{-2}$ | $2 \cdot 10^2$ | $3.6 \cdot 10^{-3}$ | $1 \cdot 10^3$ |
| | 5 | 1295 | Out of memory! | - | $1.8 \cdot 10^{-2}$ | $2 \cdot 10^2$ | $4.2 \cdot 10^{-3}$ | $2 \cdot 10^3$ |

maximum of the Chebychev bound of theorem 2 and the MoM bound of theorem 3. The number of frequencies sampled is $3.2 \cdot 10^7$.

**Comparison with TSSOS on the Chebychev basis.** We compare GloptiNets with TSSOS on random Chebychev polynomials in table 2, similarly to the comparison with trigonometric polynomials in table 1. Minimizing polynomials defined on the canonical basis is easier: contrary to trigonometric polynomials, there is no need to account for the imaginary part of the variable. If $d$ is the dimension, complex polynomials are encoded in a variable of dimension $2d$ in TSSOS, following the definition of Hermitian Sum-of-Squares introduced in Josz and Molzahn [2018]. Hence, the random polynomials we consider are characterized by the dimension $d$ and their number of coefficients $n$; instead of bounding the degree, we use all the basis elements $H_\omega(\mathbf{x}) = \prod_{\ell=1}^d H_{\omega_\ell}(\mathbf{x}_\ell)$ for which $\|\omega\|_\infty \leq p$. The maximum degree is then $dp$. The RKHS norm of $f$ is fixed to 1. As with the comparison on Trigonometric polynomial table 1, we see that GloptiNets provides similar certificates no matter the number of coefficients in $f$. Even though it lags behind TSSOS for small polynomials, it handles large polynomials which are intractable to TSSOS. The "small" and "big" models have the same structure as for the trigonometric polynomials experiments.

**Sampling from the Bessel distribution.** The function $\omega \mapsto e^{-s} I_\omega(s)$ decays rapidly. In fact, with $s = 2$, which is the value used to generate the random polynomials, it falls under machine precision as soon as $\omega > 14$. Thus, we approximate the distribution with a discrete one with weights $I_\omega(s)$ for $\omega$ s.t. the result is above the machine precision. We then extend it to multiple dimension with a tensor product. Finally, we use a hash table to store the already sampled frequency, to make the evaluation of million of frequencies much faster. For instance in dimension 5, sampling $10^6$ frequencies from the Bessel distribution of parameter $s = 2$ on $\mathbb{N}^5$ yields only $\approx 10^4$ unique frequencies. This allows for tighter certificates, as it makes the r.h.s of eq. (9), in $1/N$, much smaller. Note that the time to generate this hash table is *not* reported in tables 1 and 2, and of the order of a few seconds.

**Optimizing a kernel mixture.** As it is the case with polynomials, when optimizing a function of the form $h(x) = \sum_{i=1}^n \alpha_i K(x_i, x)$ the certificate provided by GloptiNets only depends on the function norm $\|h\|_\mathcal{H}^2$ and not on *e.g.* the number of coefficients $n$. This is illustrated in fig. 4.

# D   Fourier coefficients in linear time

**Lemma 4** (Fourier coefficient of the Bessel kernel in linear time). *Let $g$ be an extended k-SoS model as in definition 1. Then, its Fourier transform can be evaluated in* linear *time in $m$ with*

$$\widehat{g}_\omega = \sum_{k=1}^r \sum_{n \in \mathbb{Z}^d} \left( \sum_{i=1}^m R_{ki} \prod_{\ell=1}^d \phi_{\ell,-}(\mathbf{z}_{i\ell})_{n_\ell} \right) \cdot \left( \sum_{i=1}^m R_{ki} \prod_{\ell=1}^d \phi_{\ell,+}(\mathbf{z}_{i\ell}) \right) \tag{32}$$

*where*

$$\forall n \in \mathbb{Z}, z \in \mathbb{T}, \ell \in [d], \ \ \phi_{\ell,\pm}(z)_n = \sqrt{q_{\ell,n}} e^{\pi \mathrm{i}(n \pm \omega_\ell)z}$$

*and $q_{.,.}(s)$ is defined with lemma 6.*

Lemma 4 provides a formula for computing $\widehat{g}_\omega$ which is linear in $m$, but which still requires numerical approximation to compute the sum on $n \in \mathbb{Z}^d$. For instance, restraining the sum to the hyperbolic

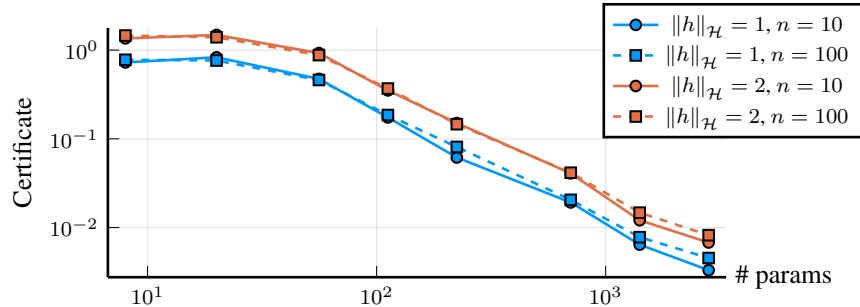

Figure 4: Certificate vs. number of parameters in $g$ when certifying mixture of Bessel functions, characterized by their RKHS norm *(1 in blue, 2 in red)* and their number of coefficients *(10 in circles, 100 in rectangles)*. As with polynomials, this shows that GloptiNets is only sensible to the former, and not to the way the function is represented. We are not aware of other algorithms able to certify this class of functions.

cross Dũng et al. [2017]

$$\mathrm{HC}(d,n) = \left\{ \omega \in \mathbb{Z}^d; \prod_{\ell=1}^{d} \max\{1, |\omega_\ell|\} \le n \right\}$$

would result in a complexity of $O(n(\log d)^n mr)$ and should produce reasonably accurate estimate of $\widehat{g}_\omega$ for low $n$.

Furthermore, since $q$ is real-even w.r.t $n$, the inner-product in eq. (36) can be simplified by computing only half of the terms.

*Proof.* From lemma 1, we have that

$$\widehat{g}_\omega = \sum_{i,j=1}^{m} R_i^\top R_j \prod_{\ell=1}^{d} e^{-2\mathbf{s}_\ell} I_{|\omega_\ell|}(2\mathbf{s}_\ell \cos \pi(\mathbf{z}_{i\ell} - \mathbf{z}_{j\ell})) e^{-\mathrm{i}\pi\omega_\ell(\mathbf{z}_{i\ell} + \mathbf{z}_{j\ell})}. \tag{33}$$

Introducing

$$f_\ell(x,y) = e^{-2\mathbf{s}_\ell} I_{|\omega_\ell|}(2\mathbf{s}_\ell \cos \pi(x-y)) e^{-\mathrm{i}\pi\omega_\ell(x+y)}, \tag{34}$$

eq. (33) simplifies to

$$\widehat{g}_\omega = \sum_{i,j=1}^{m} R_i^\top R_j \prod_{\ell=1}^{d} f_\ell(\mathbf{z}_{i\ell}, \mathbf{z}_{j\ell}). \tag{35}$$

Using lemma 6, for any $x, y \in \mathbb{T}$,

$$e^{-2\mathbf{s}_\ell} I_{|\omega_\ell|}(2\mathbf{s}_\ell \cos \pi(x-y)) = \sum_{n \in \mathbb{Z}} q_{\ell,n} e^{\pi \mathrm{i} n(x-y)}$$

($q_{\ell,n}$ depends on $\omega_\ell$) so that, $f_\ell$ defined in eq. (34) now writes

$$
\begin{aligned}
f_\ell(x,y) &= \sum_{n \in \mathbb{Z}} q_{\ell,n} e^{\pi \mathrm{i} n(x-y)} e^{-\pi \mathrm{i}\omega_\ell(x+y)} \\
&= \sum_{n \in \mathbb{Z}} q_{\ell,n} e^{\pi \mathrm{i}(n-\omega_\ell)x} e^{-\pi \mathrm{i}(n+\omega_\ell)y} \\
&= \phi_{\ell,-}(x) \cdot \phi_{\ell,+}(y)
\end{aligned}
\tag{36}
$$

where, for any $\ell \in \{1, \ldots, d\}$ and $z \in \mathbb{T}$, we defined

$$\phi_{\ell,\pm}(z) = \left( \sqrt{q_{\ell,n}} e^{\pi \mathrm{i}(n \pm \omega_\ell)z} \right)_{n \in \mathbb{Z}}. \tag{37}$$

We then define the embedding $\phi_{\pm} : \mathbb{T} \to (\mathbb{Z}^d \to \mathbb{C})$ be the tensor product of the $\phi_{\ell,\pm}$. Then, eq. (36), enables to write $\widehat{g}_\omega$ in eq. (35) as

$$
\begin{aligned}
\widehat{g}_\omega &= \sum_{i,j=1}^m \sum_{k=1}^r R_{ki} R_{kj} \phi_-(\mathbf{z}_i) \cdot \phi_+(\mathbf{z}_j) \\
&= \sum_{k=1}^r \left[ \sum_{i=1}^m R_{ki} \phi_-(\mathbf{z}_i) \right] \cdot \left[ \sum_{i=1}^m R_{ki} \phi_+(\mathbf{z}_i) \right] \\
&= \sum_{k=1}^r \left[ \sum_{i=1}^m R_{ki} \phi_{1,-}(\mathbf{z}_{i1}) \otimes \cdots \otimes \phi_{d,-}(\mathbf{z}_{id}) \right] \cdot \left[ \sum_{i=1}^m R_{ki} \phi_{1,+}(\mathbf{z}_{i1}) \otimes \cdots \otimes \phi_{d,+}(\mathbf{z}_{id}) \right] \\
&= \sum_{k=1}^r \sum_{n \in \mathbb{Z}^d} \left( \sum_{i=1}^m R_{ki} \prod_{\ell=1}^d \phi_{\ell,-}(\mathbf{z}_{i\ell})_{n_\ell} \right) \cdot \left( \sum_{i=1}^m R_{ki} \prod_{\ell=1}^d \phi_{\ell,+}(\mathbf{z}_{i\ell}) \right)
\end{aligned}
$$

which is the desired result. $\qquad \square$

# E  Other computation

**Lemma 5.** *Let $f$ be the function defined on $(-1,1)$ with*

$$
\forall u \in (0, 1/2), \quad f(\cos 2\pi u) = e^{s \cos 2\pi(u-v)}. \tag{38}
$$

*Then, its Chebychev coefficient are given with*

$$
f_\omega = (1 + \mathbf{1}_{\omega \neq 0}) \cos(2\pi \omega v) I_\omega(s). \tag{39}
$$

*Proof.* The $\omega \in \mathbb{N}_*$. The component $\omega$ of a function $f$ on the Chebychev basis is given with

$$
f_\omega = \frac{2}{\pi} \int_{-1}^1 f(x) T_\omega(x) \frac{\mathrm{d}x}{\sqrt{1-x^2}},
$$

which we conveniently rewrite, with the classical change of variable $x = \cos 2\pi u$,

$$
f_\omega = 2 \int_{I_1} f(\cos 2\pi u) \cos(2\pi \omega u) \mathrm{d}u \tag{40}
$$

which is valid for any interval $I_1 \subset \mathbb{R}$ of length 1.

Now, for $s > 0$, consider the function $f$ defined on $(-1,1)$ with $x \mapsto e^{s \cos(\arccos(x) - 2\pi v)}$, or equivalently

$$
\forall u \in (0, 1/2), \quad f(\cos 2\pi u) = e^{s \cos 2\pi(u-v)}. \tag{41}
$$

Putting eq. (41) into eq. (40), we obtain

$$
\begin{aligned}
f_\omega &= 2 \int_{I_1} e^{s \cos 2\pi(u-v)} \cos(2\pi \omega u) \mathrm{d}u \\
&= 2 \int_{I_1} e^{s \cos 2\pi u} \cos(2\pi \omega(u+v)) \mathrm{d}u \\
&= 2 \int_{I_1} e^{s \cos 2\pi u} \cos(2\pi \omega u) \cos(2\pi \omega v) \mathrm{d}u - 2 \int_{I_1} e^{s \cos 2\pi u} \sin(2\pi \omega u) \sin(2\pi \omega v) \mathrm{d}u.
\end{aligned}
$$

The last term is odd, hence integrate to 0 on an interval centered around 0. Hence,

$$
f_\omega = 2 \cos(2\pi \omega v) \int_{I_1} e^{s \cos 2\pi u} \cos(2\pi \omega u) \mathrm{d}u. \tag{42}
$$

We recognize the definition of the modified Bessel function of the first kind, defined in eq. (14). Plugging this into eq. (42), we obtain

$$
f_\omega = 2 \cos(2\pi \omega v) I_\omega(s) = 2 I_\omega(s) H_\omega(\cos(2\pi v)). \tag{43}
$$

If $\omega = 0$, we add a factor $1/2$ into the definition in eq. (40), which yields

$$
f_\omega = I_0(s). \tag{44}
$$

$\qquad \square$

**Lemma 6** (Fourier decomposition of Bessel composed with cosine). *Let $s > 0$, $\omega \in \mathbb{N}$ and $z \in \mathbb{T}$. Then,*

$$e^{-2s} I_\omega(2s \cos 2\pi z) = \sum_{n \in \mathbb{Z}} q_{\omega,n} e^{2\pi i n z},$$

$$\text{where } \forall n \geq 0, q_{\omega,n} = \begin{cases} e^{-2s} \sum_{p \geq (\frac{n-\omega}{2})_+} \frac{(s/2)^{2p+\omega}}{p!(p+\omega)!} \binom{2p+\omega}{p - \frac{n-\omega}{2}} & \text{if } n \equiv \omega, \\ 0 & \text{otherwise.} \end{cases} \tag{45}$$

*and $q_{\omega,-n} = q_{\omega,n}$ by evenness of the coefficients.*

*Proof.* From the definition of the modified Bessel function of the first kind [Watson, 1922, p.77, Eq. 2], we have

$$I_\omega(z) = \sum_{p \geq 0} \frac{(z/2)^{2p+\omega}}{p!(p+\omega)!},$$

so that

$$
\begin{aligned}
I_\omega(2s \cos 2\pi z) &= \sum_{p \geq 0} \frac{s^{2p+\omega}}{p!(p+\omega)!} \cos(2\pi z)^{2p+\omega} \\
&= \sum_{p \geq 0} \frac{(s/2)^{2p+\omega}}{p!(p+\omega)!} \left( e^{2\pi i z} + e^{-2\pi i z} \right)^{2p+\omega} \\
&= \sum_{p \geq 0} \frac{(s/2)^{2p+\omega}}{p!(p+\omega)!} \sum_{k=0}^{2p+\omega} \binom{2p+\omega}{k} e^{2\pi i (2(p-k)+\omega) z}. \tag{46}
\end{aligned}
$$

Using the change of variable $n = 2(p - k) + \omega$ into eq. (46), we see that $n$ has the same parity as $\omega$ and

$$I_\omega(2s \cos 2\pi z) = \sum_{p \geq 0} \frac{(s/2)^{2p+\omega}}{p!(p+\omega)!} \sum_{\substack{n=-(2p-\omega) \\ n \equiv \omega}}^{2p+\omega} \binom{2p+\omega}{p - \frac{n-\omega}{2}} e^{2\pi i n z}. \tag{47}$$

Equation (47) can be rewritten

$$I_\omega(2s \cos 2\pi z) = \sum_{\substack{n \in \mathbb{Z} \\ n \equiv \omega}} e^{2\pi i n z} \sum_{p \geq 0} \frac{(s/2)^{2p+\omega}}{p!(p+\omega)!} \binom{2p+\omega}{p - \frac{n-\omega}{2}} \mathbf{1}_{-(2p+\omega) \leq n \leq 2p+\omega},$$

for which eq. (45) is a concise rewriting. $\qquad \square$

