# OpenReview forum: "GloptiNets: Scalable Non-Convex Optimization with Certificates"
_NeurIPS.cc/2023/Conference — NeurIPS 2023 spotlight_

### Official Review · Reviewer_qa97 · 2023-06-15

**Soundness:** 3 good
**Presentation:** 3 good
**Contribution:** 3 good
**Rating:** 6
**Confidence:** 4

**Summary:**

This paper focuses on non-convex global optimization on the hypercube.
The approach is built on top of the framework [7] and relies on non-negativity certificates that are not only restricted to non-negative polynomials since it is applicable to any function with computable Fourier coefficients.
The methodology is flexible in the sense that one can increase the certificate of the obtained certificate by relying on the so-called k-SOS method. The approach is naturally compatible with GPU computation thanks to a modular Bessel kernel, which is stable by product. The resulting optimization method, called GloptiNets, can be handled with automatic differentiation. Afterwards the algorithm is benchmarked against the TSSOS library designed for sparse polynomial optimization on six problems of dimension 3 or 4. The performance are similar or less good when the number of coefficients is small, and certificates are obtained with GloptiNets on examples for which the SDP solver used by TSSOS runs out of memory.



**Strengths:**

The methodology allows a certain flexibility as it is not restricted to objective polynomial functions and does not require any assumptions on the structure (e.g., sparsity, symmetry) apart from the hypercube constraint.
One can also increase the accuracy of the certificate by considering bigger nets.


**Weaknesses:**

Some technical explanation are not easy to follow (see the questions below).

The title of the paper is slightly misleading, I suggest to emphasize that the optimization is performed on the hypercube.

The paper only provides numerical comparisons for low dimensions (the number of variables is either 3 or 4) and the corresponding benchmarks are cooked by hand (called synthetic in the conclusion), thus do not correspond to any concrete optimization problem.  Even if it is promising, I believe that this methodology in its current state is not yet convincing for optimizers interested in either academic or industrial applications.

In addition, the accuracy of the obtained certificate is low compared to interior-point methods. What would be the amount of computational time required to obtain a similar accuracy with GN? On Figure 2, the number of parameters required to go from 0.1 to 0.02 increases rather quickly. This can also be concluded from the numbers in Table 1. For problem 3, it is approximately 3 times slower to double the accuracy of the resulting certificate (going from 1.3e-2 to 6.9e-3). So one could believe that it would take a rather considerable amount of time to reach an accuracy of 1e-8.

I believe that the statement "TSSOS is not guaranteed to converge to f* but executes faster, and thus is on an equal footing with GloptiNets" is quite misleading for two reasons:

1) The efficiency of TSSOS increases when the optimization problem involves sparser polynomials (with low n), which is not the case here for problems 3 and 6.

2) The efficiency of TSSOS relies on the one of the tool used to solve the SDP. An alternative SDP solver such as COSMO would be certainly much more efficient but would also yield certificates with lower accuracy.

Similarly the sentence "poly-SoS methods (whose complexity scales dramatically and in a rigid way with dimension and degree of the polynomial)" is misleading. For a given certificate accuracy, there is no established estimate showing that the complexity of poly-SOS methods is higher.
So I strongly suggest to the authors to either reformulate this statement or to justify it with rigorously established statements.
The rigidity statement is also not really accurate as any structure-exploiting variant of the Moment-SOS hierarchy comes with a specific computational cost and convergence rates.

There is a potentially high source of conservativeness for the resulting lower bound because of the difference between the L-infinity and Fourier norms.

**Questions:**

l123: each z_i is a vector of T^d so shouldn't we have Z \subset T^d?

l124: which norm is used?

After l154: the matrix R suddenly (re-)introduced in Equation (7) is rather confusing. Where does this matrix come from? I guess that it is related to R from definition (1).

Figure 1 is not mentioned in the text and I could not understand it. The F norm color is displayed in orange, not in red.

(8) The expectation notation differs from (7) as there is no sub-index anymore.

In Figure 2, I cannot see any red.

Could one obtain estimates of the difference between the L-infinity and Fourier norms? Is there a specific class of examples where this difference is significant?

**Limitations:**

There is a clear section dedicated to limitations with an explicit list:
- an extensive hyper-parameter search is required
- the structure is not exploited
- the setting is restricted to the hypercube
- the benchmarks are somehow artificial

To be more convincing, the framework should be adapted to take into account the structure of real-world problems, e.g., the AC-OPF problem mentioned by the authors.

---

> ### Author Rebuttal · Authors · 2023-08-08
>
> ## About the quality of the certificate
>
> We refer to the answer to all reviewers where this matter is discussed extensively with new experiments.
>
> ## Comparison with TSSOS
>
> > The efficiency of TSSOS increases when the optimization problem involves sparser polynomials (with low n), which is not the case here for problems 3 and 6.
>
> This is indeed the point we make. We show that TSSOS is very efficient when the number of coefficients in the polynomials is small, resembling a sparse polynomial (xps 1, 2, 4, 5). However, we show that the added value of our algorithm lies on certifying dense problems, of which xps 3 and 6 are instances of.
>
> We recall this limitation in the "Limitations" section, "our model is not competitive on problems which exhibit some algebraic structure, as for instance term sparsity" (l. 306).
>
> > The efficiency of TSSOS relies on the one of the tool used to solve the SDP. An alternative SDP solver such as COSMO would be certainly much more efficient but would also yield certificates with lower accuracy.
>
> We thank the reviewer for pointing this out. We will add it to the main text.
>
> However, this does not change the conclusion of the experiments: polynomial hierarchies require forming the moment matrix, whose size scales exponentially in the number of coefficients or in the problem's dimension. Term and correlative sparsity leveraged by TSSOS mitigate this issue on sparse polynomials by turning a high-dimensional problem into multiple small-dimensional problems. However, this is not applicable in the "dense polynomials" settings we consider in xps 3 and 6. For these ones, allocating enough memory to assemble the moment matrix is not possible, so using COSMO instead of MOSEK will not change anything in those cases.
>
> ## Quality of the $F$-norm
>
> This is indeed a source of conservatism which is hard to quantify. Note however that this is the smallest norm for the norms given by summable, positive operators (Lemma 3 in [1]).
>
> [1] Blake Woodworth, Francis Bach, and Alessandro Rudi. Non-Convex Optimization with Certificates and Fast Rates Through Kernel Sums of Squares. In Proceedings of Thirty Fifth Conference on Learning Theory, pages 4620–4642. PMLR, June 2022.
>
> ## Complexity of TSSOS
>
> We will reformulate this point:
> - Given a polynomial $f$ and a tolerance $\epsilon$, neither poly-SoS nor GloptiNets provides a priori guarantees on the time complexity for certifying the positivity of $f$ up to an error $\epsilon$.
> - However, GloptiNets provides a bound on the complexity of the algorithm (as choosing the model size and the training time is left to the user), whereas poly-SoS, in their standard variant, requires a relaxation order of at least $p_{\min} = ⌈deg(f)/2⌉$, which requires assembling a (potentially block-diagonal) moment matrix of size $O(\binom{d + p_{\min}}{d}) = O(p^d)$.
>
>
> ## Answers to questions
>
> We thank the reviewers for spotting the following typos we will fix in the main text.
>
> l123: Indeed, this is $\mathbf{Z} \subset \mathbb{T}^d$ rather than $\mathbf{Z} \in \mathbb{T}^d$
>
> l124: This is the Euclidean norm, $\lVert R K_\mathbf{Z}(\mathbf{x}) \rVert_2^2$
>
> After l154: $g$ is a K-SoS model introduced in the Definition 1 above.
>
> Fig. 1 shows on a random example the possible discrepancy between the $L_\infty$ norm (what we want to estimate in blue), the $F$-norm (what is tractable to estimate, in orange) and its tractable-to-evaluate probabilistic estimate, in green. This figure is updated in the document attached to the answer to all reviewers.

---

> > ### Comment · Reviewer_qa97 · 2023-08-17
> >
> > Thank you for your responses, I will retain my score.

---

### Official Review · Reviewer_rJQd · 2023-07-05

**Soundness:** 3 good
**Presentation:** 3 good
**Contribution:** 3 good
**Rating:** 7
**Confidence:** 4

**Summary:**

This paper presents GloptiNets, a method to bound the suboptimality of a given candidate solution to the optimization of a smooth (nonconvex) function on the hypercube.

Nonconvex optimization with optimality certificates is difficult. Oftentimes it is relatively easy (fast) to compute a candidate solution, but it is much more difficult (computationally expensive) to bound the suboptimality of the solution. A popular way to provide optimality certificates is via polynomial sums of squares (Lasserre's hierarchy), but it (a) can only handle functions that are polynomials and (b) scales poorly to the dimension of the problem due to computational bottlenecks in semidefinite programming.

GloptiNets tries to tackle the drawbacks of poly-SOS. The key observation is that, given a candidate solution, any positive function provides a lower bound, and hence a bound on the suboptimality. With this key observation, and considering the recent success of machine learning, the overall idea of GloptiNets is to seek a positive function parametrized by "neural networks" (more precisely the anchor points $z$ and the matrix $R$ in Definition 1.)

The rest of the paper mostly focuses on choosing the Kernel function and leveraging sampling (and concentration inequalities) to evaluate the loss function more efficiently (for which the reviewer did not check carefully).

The experiments of this paper compare GloptiNets with TSSOS, a state-of-the-art polynomial SOS toolbox, on computing certificates for random positive trigonometric polynomials. The experiments show that although TSSOS can obtain high-accuracy certificates, GloptiNets is much more scalable.



**Strengths:**

- The paper tackles the important problem of nonconvex optimization with certificates.
- The proposed GloptiNets mitigates the drawbacks of existing poly-SOS frameworks, i.e., it can handle non-polynomial problems and it does not rely on semidefinite programming. As a result, it is more scalable.

**Weaknesses:**

The biggest issue with this paper is that (i) its experiments are rather limited (I believe there are only 6 problems tested), and (ii) the limited experiments are not even related to applications where optimality certificates are desired. Therefore, it is questionable how useful this technique will be in practice (and it kind of defeats the purpose stated in the introduction).

I would encourage more experiments where the optimality certificates are indeed valuable (for example recently in [robotics](https://arxiv.org/abs/2109.03349)).

**Questions:**

I notice that in the results table, the best suboptimality GloptiNets can produce is in the order of $10^{-3}$. I wonder if this can be improved by, for example
- increase the number of samples $x_1,\dots, x_N$,
- increase the size of the neural network (Fig. 2 partially answers this, but the accuracy there is still far from, say $10^{-9}$ as can be achieved by TSSOS)
- let the neural networks train for longer time
- increase the number of samples $\omega_1,\dots,\omega_n$

If it is possible to achieve the same certificate accuracy as TSSOS, then it is useful to report the number of sample points and training time required to attain the accuracy; if it is not possible, then an explanation of this should be provided (because the positive function class of GloptiNets should subsume the positive function class of TSSOS, and therefore the NN should be able to achieve the same accuracy as TSSOS if enough parameters are allowed?).

**Limitations:**

Limitations are clearly stated.

---

> ### Author Rebuttal · Authors · 2023-08-08
>
>
> ## Use of GloptiNets in practice
>
> Synthetic datasets offer the advantage of precise parameter control, including the function's norm and the number of coefficients for polynomials. This allows us to determine that the quality of GloptiNets certificates depends on the former and not the latter.
>
> It would be very interesting to apply GloptiNets to the example the reviewer gives in robotics; however, their model uses specific functions which exhibit discontinuities: this may require a separate focused effort for efficient certification.
>
> Moreover, while certifying polynomials is a natural application of GloptiNets, its capabilities extend beyond this specific function class. In our response to reviewers, we present new examples of successfully certifying kernel mixtures, for which GloptiNets is the only alternative we are aware of. Kernel mixtures, which represent functions learned by various kernel algorithms, are widely used in the machine learning domain, suggesting that GloptiNets could find substantial applicability there.
>
> To be more specific on this last point, given some samples $(x_i, y_i)_{1 \leq i \leq n}$ from a black-box process, one can fit a model $h(x) = \sum\_{i=1}^n \alpha_i K(x_i, x)$, where $\alpha$ is given by *e.g.* $\alpha = (K + \lambda I)^{-1} y$ for Ridge regression. Then, GloptiNets can certify the minimum of this model, which acts as a certificate on the black-box process with some additional statistical assumptions.
>
> Overall, our work serves as a proof of concept, laying the groundwork for future research to build upon and apply this framework to certify real-world applications.
>
>
> ## Tightness of the certificates
>
> We give a detailed response in the general answer to all reviewers. In there, we detail the influence of all the parameters you mentioned. In a nutshell, increasing the number of samples $x_1, \dots, x_n$ and the training time will result in better optimization, hence lower estimation error; increasing the size of the network will lower the approximation error; finally, increasing the number of frequencies $\omega_1, \dots, \omega_n$ will result in lower variance.

---

> > ### Comment · Reviewer_rJQd · 2023-08-18
> >
> > Thanks for the authors' response.
> >
> > The new experimental results are very interesting, and the new certificates obtained by L-BFGS are pretty good.
> >
> > I raised my score to Accept.

---

### Official Review · Reviewer_RR26 · 2023-07-06

**Soundness:** 4 excellent
**Presentation:** 4 excellent
**Contribution:** 3 good
**Rating:** 6
**Confidence:** 2

**Summary:**

This paper introduces a method for certain non-convex optimization problems on a hypercube or torus (i.e. with periodic boundary conditions), which also provides a certificate (a measure of how close the fitted function is to the target).
As I understand it, the certificate computation utilizes the Fourier basis.
They do two sets of experiments using their model GloptiNet: one on random trigonometric and one on Chebyschev polynomials, with increasing number of coefficients. They compare their method's performance against an existing polynomial solver, TSSOS. Their model's certificates are orders of magnitude less tight (~$10^{-2}$) than TSSOS ($10^{-12}$). But their model has the benefit that it takes almost constant time, independent of the complexity of the polynomial, to get within the certificate's accuracy. They also show that the certificate gets tighter as they increase the number of model parameters, though the drop seems slower than linear.


**Strengths:**

The paper gives a detailed exposition of the derivation of the certificate and discusses existing work extensively. I am no expert in this field, but I could more or less follow the process.
The contribution of the work, in addition to providing a certificate, seems to lie in its efficiency. They claim it should work well in higher dimensions, too, and the run time seems to be not affected much by the complexity of the target function.
Also, being able to use fast stochastic optimizations of DNN while still providing a certificate for the fit seems to be another benefit of the method.
Since I am not familiar with this branch of optimization, I cannot assess the novelty that much.


**Weaknesses:**

A big emphasis is on the derivation of the certificate. While I appreciate the details, it makes it difficult to follow the flow of the paper. Instead a little more details on how the extended k-SoS model is implemented and how the optimization is done would be useful. I see discussions of the details in the appendix, but it is brushed over in the main text. For example, is the R matrix optimized, or fixed?
Also, the high value of the certificate and lack of hyperparameter tuning, except model size, weakens the paper (see questions).

Additionally, the target function in the two experiment, trigonometric and Chebyshev, are both related to the Fourier series, which is used in computing the certificate. I wonder how well the model performs on other polynomials with other bases (see questions).


**Questions:**

1. currently the certificate seems very loose at $O(10^{-2})$. Is this a good number?
2. How much does hyperparameter tuning (other than model size) affect the certificate? Is there hope for much better values, or does it need dramatic changes?
3. Experiments: How would the model perform on polynomials from a dramatically different basis from the Fourier basis (e.g. $x^p$)? I understand this doesn't work on the Torus (no periodicity), but does the hypercube have a similar limitation (meaning you *have to* use the Chebyshev basis due to boundary conditions)?
4. Baseline methods seem very limited. Is TSSOS the only viable alternative?
5. In Fig 2, no red curve is presented, but mentioned in the caption.


**Limitations:**

They discuss the limitations. I would also add discussions about the class of polynomials, if it is currently limited to Chebyshev and trigonometric polynomials. As for societal impact, I don't see any issues, as this is a purely mathematical optimization paper.

---

> ### Author Rebuttal · Authors · 2023-08-08
>
> ## Quality of the certificate
>
> We refer the reviewer to the common response on that aspect.
>
> ## Implementation details
>
> We would be happy to include implementation details in the main text, as a lot of attention was dedicated to having a structure which can scale to thousands of parameters.
>
> For the optimization, we used gradient descent with momentum and a cosine scheduler. This was the most classical configuration we found in the DL literature. We performed the  hyperparameter search on the type of regularization (proxy for the variance or not), the value of the regularization and the learning rate.
>
> Finally, to answer precisely to your question, both the coefficients $R$ and the anchor points $\mathbf{z}$ are optimized during the training (`for` loop in Algorithm 1).
>
> ## Other polynomial basis
>
> Simply put, for polynomial optimization there are no limitations on the basis. Indeed, if the polynomial $h$ is given in another real (resp. complex) basis, we can first perform a change of basis to the Chebychev (resp. Fourier) basis, which is a linear operation. The resulting polynomial would  be handled natively by GloptiNets. More than that, the algorithm applies to the minimization of any function defined on a hypercube $C = [a_i, b_i]_{1 \leq i \leq n}$, for which we have access to the Fourier/Chebychev coefficients.
>
> Here is a more detailed response:
>
> **Hypercube constraints.**
> Given a function $\tilde{h}$, what the algorithm requires is a set of constraints $C = [a_i, b_i]_{1 \leq i \leq n}$ to localize the minimum. Lassere's polynomial hierarchies do not necessarily need such constraint, but have much better convergence guarantees under similar constraints. For instance, this happens if $\lVert x \rVert^2 \leq R$ is in the constraint set, which is called the "Archimedean property"  – this is encompassed in the hypercube constraint $C$ we consider. Thus, let's assume we minimize $h(x) = \tilde{h}(a_i + (x - a_i) / (b_i - a_i))$ for $x \in [0, 1]^d$.
>
> **No loss of generality with periodic functions**.
> Now, define $f(x) = h(\cos(2\pi u))$. This is a $1$-periodic function, **whose minimum is the same as $h$**. This is detailed in Remark 1, l. 109. This shows that working with trigonometric polynomials is done with no loss of generality.
>
> **Better implementation with the Chebychev basis.**
> One caveat on the transformation $f(x) = h(\cos(2\pi u))$ is that $f$ is an even function in all dimensions: GloptiNets with Fourier series would minimize $f$ on $(0, 1)^d$ whereas looking at $(0, ½)^d$ would be sufficient! Hopefully, we can overcome this bad dependency with the dimension by working directly with Chebychev series. They form an orthonormal basis of (non-periodic) functions defined on $(-1, 1)$. Assume we have access to the coefficients of $h$ on this basis, i.e. $h(x) = \sum h_\omega T_\omega(x)$ where $T_\omega$ is the Chebychev polynomial of degree $\omega$. Then, the key property we leverage is that $|T_\omega(x)| \leq 1$ on $(-1, 1)$. This implies that
> $$
> \lVert h \rVert_\infty \leq \sum_{\omega} |h_\omega| := \lVert h \rVert_F, ~ \text{which allows} ~ h_\star \geq c - \lVert h - c - g \rVert_F, ~~ \forall c \in \mathbb{R}, g \geq 0.
> $$
> This is exactly Eq. $(2)$ l.111, which allows for the same analysis as the one with the Fourier series. Note that we could have the same relation with the canonical basis you mentioned (i.e. $|x^p| \leq 1$ on $(-1, 1)$) but this would likely result in a big gap between the resulting $F$-norm and the $L_\infty$ norm.
>
>
> To conclude, an algorithm which handle Fourier series would be enough. However, we obtain a more efficient implementation with the Chebychev basis, for which any polynomials can be converted to.
>
>
> ## Other solvers
>
> We compared to TSSOS as it is the state of the art for fast (approximate) certification of polynomials. On top of that, it handles both complex and real polynomial hierarchies, with well-functioning implementation available.

---

> > ### Comment · Reviewer_RR26 · 2023-08-16
> > **Thanks for the reply**
> >
> > Thank you for clarifying the points. I guess for most problems of interest the bases you used are general enough. And thank you for the new experiments. The choice of optimizer seems to impact the certificate value significantly.

---

### Official Review · Reviewer_Sbjx · 2023-07-09

**Soundness:** 4 excellent
**Presentation:** 4 excellent
**Contribution:** 3 good
**Rating:** 8
**Confidence:** 4

**Summary:**

This paper develops an approach to compute certificates for non-convex optimization on functions that are optimized over the hypercube or torus. It does so by considering the generic certificate recipe from [7] and then relaxing the positive semi-definite constraint to a class of functions that is easier to optimize that they denote K-SoS. This class of functions depends on the choice of a kernel K, of which they explore different alternatives. The authors consider optimization aspects of this general framework and provide a concrete algorithm that they name GloptiNets. They also provide experiments comparing the tightness of the certificate across the number of parameters and related approaches.

**Strengths:**

1. The paper is clear and beautifully written. It is a pleasure to read.

2. Results are sound to the best of my knowledge. The theoretical derivations are clear and give the proper references to understand which steps have been taken.

3. The authors consider all relevant aspects of the implement of this framework: from expressiveness properties of the function approximation, to probabilistic estimates of the certificate, to practical implementation details.

**Weaknesses:**


No major weaknesses identified. Below are some minor suggestions that I hope the authors consider, but should not be taken as criticism towards the work.


## Minor suggestions

1. Add x and y labels to Figure 1.
2. Some aspects of this paper, such as the lower bound formula (2) or the anchor points Z are introduced assuming the reader knows about the related works. I would encourage the authors to aim for a larger audience. In particular, a few words about the intuition can go a long way towards making the work more accessible.

## Typos

L244: eventhough -> even though
L245 optimisation -> optimization (since American spelling is used elsewhere)
L93: "we focus on the computational performances of our model" it's unclear what the authors mean here by computational performances, is it run-time? tightness of the certificate? the trade-off between these two? Please make precise

**Questions:**

1. What is Z_d in eq. (1)? I don't think it has been defined before.

2. It was not clear to me whether the anchor points Z are given by the problem or are part of the choice of the kernel family. Clarification would be appreciated.



**Limitations:**

I believe the limitations have been correctly addressed by the author in the Limitations section.

---

> ### Author Rebuttal · Authors · 2023-08-08
>
> We thank this reviewer for the kind and encouraging feedback!
>
> As you suggest, we will use additional space to provide additional assumption on the derivation of the certificate in Eq. (2). Typically, it relies on the relation
> $$
> f_\star = \sup_c c ~~ \text{ s.t. } ~~ f - c \geq 0 ~~~~~~~~ \text{(A)}
> $$
> which is a convex problem, albeit with a dense set of constraint. From there, a penalized version of $(A)$ is given with
> $$
> f_\star = \sup_{c, g \geq 0} c - \lVert f - c - g \rVert_\infty ~~~~~ (B)
> $$
> introduced in [1], which is still untractable (this actually highlights the fact that approximating $f$ is as hard as finding $f_\star$) but provides a general recipe for computing certificates, by taking any family of positive function $g$ (here, kernel-SoS models), and any surrogate for the $L_\infty$ norm (here, the F-norm). This yields Eq. $(2)$ in the paper. Adding an efficient way to evaluate it (as the $F$-norm is an infinite sum) and optimize it is the gist of our contribution.
>
> ## Questions
>
> $\mathbb{Z}$ is the set of integers, which was indeed not introduced before.
>
> Our positive function class are functions of the form $x \mapsto {\bf v}(x)^\top G {\bf v}(x)$, with $G \succeq 0$ and ${\bf v}(x) = (v_i(x))_{1 \leq i \leq n}$. We realized that optimizing the embedding ${\bf v}$ is beneficial for the certificate; that is why we take ${\bf v}$ of the form $v_i(x) = K(z_i, x)$. The setup we tried is $K$ set to the Bessel kernel (l. 217 or Eq. $(14)$) and $z_i \in \mathbb{T}^d$ the anchor point tuned during the optimization. In a kernel-learning vocabulary, this is learning the anchor points of a Nyström projection.
>
>
> [1] Blake Woodworth, Francis Bach, and Alessandro Rudi. Non-Convex Optimization with Certificates and Fast Rates Through Kernel Sums of Squares. In Proceedings of Thirty Fifth Conference on Learning Theory, pages 4620–4642. PMLR, June 2022.

---

> > ### Comment · Reviewer_Sbjx · 2023-08-20
> >
> > I thank the authors for their clarification answers. I keep my score unchanged.

---

### Official Review · Reviewer_rug7 · 2023-07-21

**Soundness:** 3 good
**Presentation:** 3 good
**Contribution:** 3 good
**Rating:** 6
**Confidence:** 1

**Summary:**

They present a novel approach to non-convex optimization with certificates, which handles smooth functions on the hypercube or on the torus. Unlike traditional methods that rely on algebraic properties, our algorithm exploits the regularity of the target function intrinsic in the decay of its Fourier spectrum.

**Strengths:**

The paper is clearly written and the theory is interesting.

**Weaknesses:**

The experimental results are not abundant.

**Questions:**

No.

**Limitations:**

No.

---

> ### Author Rebuttal · Authors · 2023-08-08
>
> We direct the reviewer's attention to the new experiments presented in the collective response to all reviewers. Should they have any additional concerns or questions, we would be happy to discuss them further.

---

> > ### Comment · Reviewer_rug7 · 2023-08-20
> > **Thanks for the detailed rebuttal.**
> >
> > Thanks for the detailed rebuttal. It solved my concerns. I will keep my score.

---

### Author Rebuttal · Authors · 2023-08-08

We thank the reviewers for their time and helpful feedback. Individually, we address each of their questions. Furthermore, we showcase new experiments demonstrating two key points: (1) the ability to obtain tighter certificates than the ones reported in the paper and (2) the versatility of our framework beyond polynomial optimization.

## Second-order optimization for tighter certificates

### Motivation

As pointed out by reviewers `RR26`, `rJQd` and `qa97` and detailed in Sec. 5 "Limitations", the certificates obtained by GloptiNets on **small polynomial** optimization task are not as tight as the ones obtained by polynomial solvers (Table 1, lines 1-2,4-5).

The certificates we compute are of the form `Fnorm_approx + var/√N`, where `N` is the number of frequencies sampled. The second term can be made negligible by choosing $N \gg 1$. The first term measures the quality of the approximation of $f - f_\star$ by out extended k-SoS model $g_\theta$, where $\theta = (R, \mathbf{z})$ as in Definition 1. More specifically, it is the sum of the approximation error and the estimation error. Given the fact that k-SoS models are universal approximators, the former will go to $0$ if we add more parameters to $g_\theta$. To make the latter smaller, we perform **approximate second-order optimization** in Algorithm 1 with L-BFGS.

### Results

The results for these experiments are available in **Table A** in the new material attached.

On the bigger model, using L-BFGS enables the bigger model (GN-big) to have a certificate about **one order of magnitude** tighter that the ones presented in the paper. The smaller model also benefits from this procedure, altough marginally. This shows that in the experiments reported in Table 1 in the original submission,  GN-big suffered from a high estimation error.

To conclude, tighter certificates than the one presented in Table 1 are achievable by GloptiNets. This requires (1) sampling enough frequencies $N \gg 1$ *(to lower the variance term)*, (2) large enough models *(for the approximation error)*, and (3) good enough optimization *(for the estimation error)*. We detailed the first two points in our paper, but will definitely add the third in the discussion of the experiment section.

We thank the reviewers who motivated these new results.

## Optimizing Kernel mixture

### Motivation

Although we compared our approach to TSSOS for certifying polynomials, our model is not confined to this function class. This is evidenced by successful experiments on kernel mixtures, where our approach stands as the only viable alternative we are aware of.

In our submission, we carried out experiments with polynomials so that we could compare with existing solvers. The conclusion was that GloptiNets was no match to TSSOS when certifying small polynomials $f$ which exhibits some algebraic structure, but had a complexity independent of the number of coefficients of $f$. This enables GloptiNets to certify polynomials which are out of reach of TSSOS (Table 1, lines 3,5). Here, we go one step further and optimize a kernel mixture, which **has no polynomial structure**.

### Results

The results are showcased in **Figure A** in the document attached.

The function we certify are of the form $f(x) = \sum_{i=1}^n \alpha_i K(x_i, x)$, where $K$ is the Bessel kernel. Such function are ubiquitous in machine learning and arise *e.g.* when performing kernel ridge regression. They are characterized by their number of coefficients $n$ and their RKHS norm $\lVert f \rVert_{\mathcal{H}_s}^2$. Following Algorithm 1 as outlined in the paper, we use GloptiNets and plot the certificate function of the network size.

We can draw the same conclusion as seen in Figure 2 in the paper.
Firstly, GloptiNets' certificates depend solely on the function's norm, independent of its representation with the number of basis functions $K(x_i, x)$. Secondly, bigger networks provide tighter certificates.

Thus, even though the criticism that GloptiNets is not as tight as TSSOS is fair, it is only valid when certifying **small polynomials**. For the new showcased experiments, GloptiNets is the only alternative we are aware of.

## Update Fig. 1

We update Fig. 1 from the paper to add error bars on the random realizations and improve the rendering.

---

### Decision · Program_Chairs · 2023-09-21

**Decision:**

Accept (spotlight)

**Comment:**

This paper studies non-convex optimization with certificates, where the output of the algorithm and the estimate minimizer are provided and an explicit precision is constituted by the approach presented in this paper. This paper is theoretically solid. All the reviewers are satisfied with this paper and the revision. I suggest to accept it as a spotlight. The AC had a thorough discussion with the SAC and the SAC agreed on the decision.